# Dual Natural Gradient Descent for Scalable Training of Physics-Informed Neural Networks

**Anas Jnini**                                                                    *anas.jnini@unitn.it*
*Department of Information Engineering and Computer Science*
*University of Trento, Trento, Italy*
**Flavio Vella**                                                                  *flavio.vella@unitn.it*
*Department of Information Engineering and Computer Science*
*University of Trento, Trento, Italy*

**Reviewed on OpenReview:** *https://openreview.net/forum?id=GDHVRy6SDd*

## Abstract

Natural–gradient methods markedly accelerate the training of Physics-Informed Networks (PINNs), yet their Gauss–Newton update must be solved in the *parameter space*, incurring a prohibitive $\mathcal{O}(n^3)$ time complexity, where $n$ is the number of network trainable weights. We show that exactly the same step can instead be formulated in a *generally smaller residual space* of size $m = \sum_\gamma N_\gamma d_\gamma$, where each residual class $\gamma$ (e.g. PDE interior, boundary, initial data) contributes $N_\gamma$ collocation points of output dimension $d_\gamma$.

Building on this insight, we introduce *Dual Natural Gradient Descent* (D-NGD). D-NGD computes the Gauss–Newton step in residual space, augments it with a geodesic-acceleration correction at negligible extra cost, and provides both a dense direct solver for modest $m$ and a Nyström-preconditioned conjugate-gradient solver for larger $m$.

Experimentally, D-NGD scales second-order PINN optimization to networks with up to 12.8 million parameters, delivers one- to three-order-of-magnitude lower final error $L^2$ than first-order (Adam, SGD) and quasi-Newton methods, and —crucially —enables full natural gradient training of PINNs at this scale on a single GPU.

## 1 Introduction

**Partial Differential Equations (PDEs)**  Partial Differential Equations (PDEs) form the backbone of mathematical models used to describe a wide array of physical phenomena—ranging from fluid flow and heat transfer to the behavior of advanced materials. Conventional discretization-based techniques, such as finite element and spectral methods, often demand highly refined meshes or basis expansions to attain the desired level of accuracy. This refinement drives up computational costs, especially in engineering scenarios that call for numerous simulations with varying boundary conditions or parameter sets. In recent years, machine learning approaches—most notably those employing neural networks—have emerged as a promising complement or substitute for these traditional solvers, offering potential gains in efficiency and flexibility Raissi et al. (2019); Li et al. (2021); Jnini & Vella.

**Physics-Informed Neural Networks (PINNs)**  PINNs are a machine learning tool to solve forward and inverse problems involving partial differential equations (PDEs) using a neural network ansatz. They have been proposed as early as Dissanayake & Phan-Thien (1994) and were later popularized by the works Raissi et al. (2019); Karniadakis et al. (2021). PINNs are a meshfree method designed for the seamless integration of data and physics. Applications include fluid dynamics Cai et al. (2021); Jnini et al. (2025a;b), solid mechanics Haghighat et al. (2021) and high-dimensional PDEs Hu et al. (2023) to name but a few areas of ongoing research.

Despite their popularity, PINNs are notoriously difficult to optimize Wang et al. (2020) and fail to provide satisfactory accuracy when trained with first-order methods, even for simple problems Zeng et al. (2022); Müller & Zeinhofer (2023). Recently, second-order methods that use the function space geometry to design gradient preconditioners have shown remarkable promise in addressing the training difficulties of PINNs Zeng et al. (2022); Müller & Zeinhofer (2023); Ryck et al. (2024); Jnini et al. (2024); Müller & Zeinhofer (2024). While second-order optimizers, such as those based on Gauss-Newton (GN) principles, can harness curvature information for improved convergence, their canonical forms entail $O(n^3)$ per-iteration complexity and $O(n^2)$ memory for $n$ parameters, rendering them impractical for large-scale deep neural networks. Matrix-free methods have been proposed to compute Gauss-Newton directions without explicitly forming the Hessian Martens (2010); Schraudolph (2002); Zeng et al. (2022); Jnini et al. (2024). Despite reducing computational costs, these methods suffer from ill-conditioning, leading to slow convergence for large networks without efficient preconditioners.

To address these challenges, this work proposes **Dual Natural Gradient Descent**, a novel optimization framework incorporating the following key features:

- **We propose a primal–dual viewpoint of the Gauss–Newton step:** instead of solving the usual $n$-dimensional normal equations in parameter space, we work with their dual in the $m$-dimensional residual feature space, where $m = \sum_\gamma N_\gamma d_\gamma$, with each residual class $\gamma$ (e.g. PDE interior, boundary, initial data) contributing $N_\gamma$ collocation points of dimension $d_\gamma$, and since $m \ll n$ in practical PINNs this dual system is far cheaper to assemble, store, and solve than the original parameter-space system.

- **We propose an efficient geodesic-acceleration correction within the same dual framework.** A second-order (geodesic) term is obtained by solving one additional linear system with the same left-hand side and reuses the factorisation already built for the primary dual operator, adding negligible overhead while improving step quality.

- **We propose a low-rank Nyström spectral preconditioner for large batch sizes.** When $m$ is too large for direct factorisation, we propose using a Hessian-Free iterative method with a preconditioned conjugate gradient, we provide an efficient Nyström preconditioner based on column sampling for the dual problem.

- **We demonstrate scalability and accuracy on several PDE benchmarks.** The resulting dual natural-gradient method trains PINNs with tens of millions of parameters and dimensions in the hundreds of thousands, consistently outperforming first-order and quasi-Newton baselines by more than an order of magnitude across several representative problems. To the best of our knowledge, our contribution is the first to extend Natural-Gradient methods to PINNs of this scale.

**Related Works**

**Second-order Optimization in PINNs**  The challenge of effectively training PINNs has spurred significant research, with a growing consensus underscoring the necessity of second-order optimization methods. Recent literature highlights this trend: approaches leveraging an infinite-dimensional perspective, for instance, have demonstrated the potential to achieve near single-precision accuracy Zeng et al. (2022); Müller & Zeinhofer (2023); Ryck et al. (2024); Jnini et al. (2024); Zampini et al. (2024); Schwencke & Furtlehner (2024); Mckay et al. (2025). However, the practical application of these methods is often constrained by their high per-iteration cubic computational cost, particularly when scaling to larger network architectures due to the need to solve substantial linear systems in the parameter space. Exploration into more scalable alternatives includes quasi-Newton methods, which Kiyani et al. (2025) evaluate for their efficiency and accuracy across stiff and non-linear PDEs by leveraging historical gradient information. Complementing this, Wang et al. (2025) provide a theoretical framework for gradient alignment in multi-objective PINN training, demonstrating how second-order information, through Hessian preconditioning, can resolve directional conflicts between different loss components, thereby demanding more sophisticated curvature-aware approaches.

**Hessian-Free Curvature Approximation** Matrix-free methods have been proposed to compute Gauss-Newton directions without explicitly forming the Hessian Martens (2010); Schraudolph (2002); Zeng et al. (2022); Jnini et al. (2024). Despite reducing computational costs, these methods suffer from ill-conditioning, leading to slow convergence for large networks without efficient preconditioning Jnini et al. (2024). Our algorithm addresses this by proposing an efficient preconditioner for the dual system, significantly improving the inner solver convergence. This idea of cutting large gaps within the leading eigenvalues of the Hessian spectrum is also aligned with recent advances in preconditioning techniques, such as volume sampling Rodomanov & Kropotov (2020), polynomial preconditioning Doikov & Rodomanov (2023), and spectral preconditioning Doikov et al. (2024); Frangella et al. (2023).

**Efficient Second-Order Methods via Primal-Dual Formalism** Our work introduces a **primal-dual formalism** to scalably solve the regularized Gauss-Newton (GN) least-squares problem that arises in PINN training. This primal-dual viewpoint, which moves the main computation from the large parameter space to a smaller dual space, has been previously linked in other fields to the **push-through identity** Henderson & Searle (1981). For instance, this connection is foundational in **kernel methods** and **Gaussian Processes**, where it is the basis for the **kernel trick**. Despite its effectiveness, this approach has remained largely unexplored within the Natural Gradient Descent (NGD) and, in particular, the PINN communities. Related algebraic techniques have appeared, for example in the work. Benzing (2022), to create a scalable step for the Fisher Information Matrix.

**Geodesic Acceleration in Natural Gradient Methods** Beyond the natural gradient step, higher-order corrections can improve convergence speed and step quality. The concept of using a geodesic acceleration term, which accounts for the curvature of the manifold along the gradient path, was notably explored by Song et al. (2018); Bonfanti et al. (2024) and can be interpreted as an approximation of the Riemannian Euler method. Our primary contribution in this context is to demonstrate that this acceleration term can be computed with negligible overhead within our dual framework. By reusing the existing factorization of the dual operator for a second linear solve, we make the geodesic correction practical and scalable, integrating it seamlessly into our optimization workflow without significant performance penalty.

**Connection to Operator Learning** While our work focuses on single-instance PDE solutions learned through PINNs, the scalability of D-NGD is also highly relevant to the more general task of *operator learning*. Architectures like Physics-Informed Neural Operators (PINOs) Li et al. (2021) and PIDeepONets Wang et al. (2021) learn mappings between function spaces and often require models with vast parameter dimensions ($n$), rendering traditional $\mathcal{O}(n^3)$ second-order methods impractical. Crucially, these frameworks incorporate a PDE residual term, creating a nonlinear least-squares objective analogous to that of standard PINNs. Because D-NGD's complexity scales with the residual dimension ($m$) rather than the parameter count ($n$), it provides a practical pathway to apply curvature-aware optimization to these large-scale models, a domain that has thus far been largely reliant on first-order methods.

## 2 Preliminaries

### 2.1 Physics-Informed Neural Networks

For a given domain $\Omega \subset \mathbb{R}^d$ (or $\Omega_T = I \times \Omega$ for time-dependent problems, where $I$ is a time interval), consider a general PDE of the form

$$\mathcal{L}u = f \quad \text{in } \Omega,$$

subject to initial and boundary conditions, collectively denoted as $u = g$ on $\partial\Omega$ (where $\partial\Omega$ here generally refers to the spatio-temporal boundary). PINNs approximate the solution $u$ of the PDE using a neural network ansatz $u_\theta$, parameterized by $\theta$. The loss function is defined as:

$$L(\theta) = \frac{1}{2N_\Omega} \sum_{n=1}^{N_\Omega} \left(\mathcal{L}u_\theta(x_n) - f(x_n)\right)^2 + \frac{1}{2N_{\partial\Omega}} \sum_{n=1}^{N_{\partial\Omega}} \left(u_\theta(x_n) - g(x_n)\right)^2, \tag{1}$$

where $\{x_n \in \Omega\}_{n=1}^{N_\Omega}$ are the collocation points in the interior of the domain and $\{x_n \in \partial\Omega\}_{n=1}^{N_{\partial\Omega}}$ are the points on which initial and boundary conditions are enforced.

## 2.2 Gauss–Newton method for PINNs

**Residuals and Jacobian.** Let $i = 1, \ldots, N_\Omega$ and $j = 1, \ldots, N_{\partial\Omega}$. Define the discrete residual map to $r : (\theta) \to \mathbb{R}^m$ to be

$$r(\theta) = \big(r_\Omega(\theta), r_{\partial\Omega}(\theta)\big) \in \mathbb{R}^m, \qquad m = N_\Omega d_\Omega + N_{\partial\Omega} d_{\partial\Omega},$$

with

$$r_\Omega(\theta)_i = \frac{1}{\sqrt{N_\Omega}}\big(\mathcal{L}\, u_\theta(x_i) - f(x_i)\big) \in \mathbb{R}^{d_\Omega}, \quad r_{\partial\Omega}(\theta)_j = \frac{1}{\sqrt{N_{\partial\Omega}}}\big(u_\theta(x_j^b) - g(x_j^b)\big) \in \mathbb{R}^{d_{\partial\Omega}}.$$

Its Jacobian is $J(\theta) = \partial_\theta r(\theta) \in \mathbb{R}^{m \times n}$ and the loss is

$$L(\theta) = \tfrac{1}{2}\|r(\theta)\|_2^2.$$

**Gauss–Newton natural gradient descent.** First-order optimizers, such as gradient descent and Adam, often fail to provide satisfactory results due to the ill-conditioning and non-convexity of the loss landscape, as well as the complexities introduced by the differential operator $\mathcal{L}$ Wang et al. (2020). Instead, function-space-inspired second-order methods have lately shown promising results Zeng et al. (2022). For the remainder of this paper, we adopt *Gauss–Newton Natural Gradient Descent* (GNNG) Jnini et al. (2024). Linearizing the residual map in function space and pulling the resulting Gauss–Newton operator onto the tangent space of the ansatz yields the Gauss–Newton Gramian:

$$G(\theta) = J(\theta)^\top J(\theta), \tag{2}$$

and update rules at iteration $k$:

$$\theta_{k+1} = \theta_k - G(\theta_k)^\dagger \, \nabla_\theta L(\theta_k), \qquad k = 0, 1, 2, \ldots. \tag{3}$$

It was shown that the Gauss–Newton direction in function space corresponds exactly to the Gauss–Newton step for $L(\theta) = \tfrac{1}{2}\|r(\theta)\|_2^2$ in parameter space when the same quadrature points are employed Jnini et al. (2024); Müller & Zeinhofer (2024).

**Least–squares characterisation.**

Equivalently, the Gauss–Newton increment is the solution of the linearized problem

$$\Delta\theta^\star \;=\; \arg\min_{\Delta\theta \in \mathbb{R}^n} \tfrac{1}{2}\big\|r(\theta_k) + J(\theta_k)\,\Delta\theta\big\|_2^2, \tag{4}$$

whose first-order optimality conditions give the normal equations

$$J(\theta_k)^\top J(\theta_k)\,\Delta\theta^\star \;=\; -\,J(\theta_k)^\top r(\theta_k), \tag{5}$$

Given that $\nabla_\theta L(\theta_k) = J(\theta_k)^\top r(\theta_k)$, we recover the update equation 3.

# 3 Optimization in the Residual Space

In this section we propose a primal–dual viewpoint on Gauss–Newton updates for PINNs: the same parameter update can be cast either as an $n \times n$ linear system in parameter space or as an $m \times m$ system in residual space (where in typical cases $m \ll n$), allowing the residual-space system—which can be solved by a dense factorization or a matrix-free iterative method—to scale to problems with large batch sizes. In Section 3.3 we detail the iterative solver and the low-rank Nyström preconditioner that accelerates it. We incorporate the classic Levenberg–Marquardt damping $\lambda > 0$ Levenberg (1944); Marquardt (1963) into the Gauss–Newton model equation 4, defining the Tikhonov-regularised problem

$$\min_{\Delta\theta_k \in \mathbb{R}^n} F_\lambda(\Delta\theta_k) := \tfrac{1}{2}\big\|r(\theta_k) + J(\theta_k)\,\Delta\theta_k\big\|_2^2 + \tfrac{\lambda}{2}\,\|\Delta\theta_k\|_2^2.$$

**Proposition 3.1** (Primal normal equations). *For any parameter vector $\theta_k \in \mathbb{R}^n$, define the* primal Gramian

$$G(\theta_k) := J(\theta_k)^\top J(\theta_k).$$

*The unique minimiser $\Delta\theta_k^\star \in \mathbb{R}^n$ of the damped least-squares subproblem then satisfies*

$$\left(G(\theta_k) + \lambda I_n\right) \Delta\theta_k^\star = -\nabla_\theta L(\theta_k).$$

*Here, $\nabla_\theta L(\theta_k) = J(\theta_k)^\top r(\theta_k)$ is the gradient of the unweighted least-squares loss $\frac{1}{2}\|r(\theta_k)\|_2^2$.*

*Proof Outline:* The proof follows by standard differentiation of the objective function $F_\lambda(\Delta\theta_k)$ with respect to $\Delta\theta_k$ and setting the result to zero. The objective is strictly convex for $\lambda > 0$, guaranteeing a unique minimizer. The detailed derivation is provided for completeness in Appendix A.1.1.

The linear system given in Proposition 3.1 corresponds to the classical Levenberg–Marquardt update in the $n$-dimensional parameter space. Forming and factorizing the matrix $J(\theta_k)^\top J(\theta_k) + \lambda I_n$ in equation 3.1 requires $\mathcal{O}(mn^2 + n^3)$ time and $\mathcal{O}(n^2)$ memory, which becomes impractical when $n$ is large.

### 3.1 Dual Formulation via KKT in Residual Space

An alternative formulation to the system equation 3.1 can be derived by working in the $m$-dimensional residual space.

**Definition 3.1** (Residual Gramian). *Let the residual Gramian matrix be defined as*

$$\mathcal{K}_k := J(\theta_k) J(\theta_k)^\top \in \mathbb{R}^{m \times m}.$$

We introduce an auxiliary variable $y_k := J(\theta_k)\Delta\theta_k \in \mathbb{R}^m$ and enforce the constraint $y_k = J(\theta_k)\Delta\theta_k$ using Lagrange multipliers $\nu_k \in \mathbb{R}^m$. The Lagrangian for the subproblem of minimizing $F_\lambda(\Delta\theta_k)$ (from Proposition 3.1) becomes

$$\mathscr{L}(\Delta\theta_k, y_k, \nu_k) = \tfrac{1}{2}\|r(\theta_k) + y_k\|_2^2 + \tfrac{\lambda}{2}\|\Delta\theta_k\|_2^2 + \nu_k^\top\left(y_k - J(\theta_k)\Delta\theta_k\right).$$

Applying the Karush–Kuhn–Tucker (KKT) conditions for optimality provides the dual formulation.

**Theorem 3.1** (Dual Normal Equations). *Let $y_k := J(\theta_k)\Delta\theta_k \in \mathbb{R}^m$ be the predicted residual decrement associated with a parameter step $\Delta\theta_k \in \mathbb{R}^n$. Applying the KKT conditions to the minimization of $F_\lambda(\Delta\theta_k)$ yields the dual linear system for the optimal $y_k^\star$:*

$$(\mathcal{K}_k + \lambda I_m)\, y_k^\star = -J(\theta_k)\,\nabla_\theta L(\theta_k). \tag{6}$$

*The parameter update can then be recovered by*

$$\Delta\theta_k^\star = -\frac{1}{\lambda}\left(J(\theta_k)^\top y_k^\star + \nabla_\theta L(\theta_k)\right). \tag{7}$$

*Conversely, any pair $(\Delta\theta_k, y_k)$ satisfying equation 6 and equation 7 constitutes the unique primal–dual optimum of the Levenberg-Marquardt subproblem.*

*Proof Outline:* The KKT conditions are derived by setting the partial derivatives of the Lagrangian $\mathscr{L}(\Delta\theta_k, y_k, \nu_k)$ with respect to $\Delta\theta_k$, $y_k$, and $\nu_k$ to zero. Algebraic manipulation of these conditions yields the dual system and the recovery formula for the parameter update. The full proof is available in Appendix A.1.2.

**Primal vs Dual Perspectives.** We have derived two equivalent formulations for the LM step: the primal system in the $n$-dimensional parameter space, involving the $n \times n$ primal Gramian $G + \lambda I_n$ (with $G = J^\top J$), and the dual system in the $m$-dimensional residual space, involving the $m \times m$ matrix $\mathcal{K}_k + \lambda I_m$ (with $\mathcal{K}_k = J(\theta_k)J(\theta_k)^\top$). In practice one solves the smaller system directly—using the primal when $n \ll m$ (cost $O(n^3)$) or the dual when $m \ll n$ (cost $O(m^3)$)—or resorts to an iterative, matrix-free Preconditioned Conjugate Gradient solver with preconditioning for large-scale problems.

### 3.2 Geodesic Acceleration

When interpreting the Levenberg–Marquardt step as a velocity $\boldsymbol{v}_k$ along a geodesic in parameter space, one can improve the update by adding a second-order correction that accounts for an acceleration $\boldsymbol{a}_k$ along that same geodesic:

$$\theta_{k+1} = \theta_k + \boldsymbol{v}_k + \tfrac{1}{2}\,\boldsymbol{a}_k,$$

where $\boldsymbol{a}_k$ satisfies

$$J(\theta_k)\,\boldsymbol{a}_k \;=\; -\,f_{vv},$$

and

$$f_{vv} := \left.\frac{d^2}{dt^2}\, r\big(\theta_k + t\,\boldsymbol{v}_k\big)\right|_{t=0} \;\in\; \mathbb{R}^m$$

is the second directional derivative of the residual along $\boldsymbol{v}_k$. Computing $f_{vv}$ involves two Jacobian-vector products, capturing curvature beyond the standard LM step. Below we show that this geodesic-acceleration term can also be computed using our proposed primal–dual formalism.

**Definition 3.2** (Geodesic-Acceleration Subproblem). *At iteration $k$, the acceleration correction $\boldsymbol{a}_k$ is the minimiser of the damped system*

$$\min_{\boldsymbol{a}\in\mathbb{R}^n}\; \tfrac{1}{2}\big\|J(\theta_k)\,\boldsymbol{a} + f_{vv}\big\|_2^2 \;+\; \tfrac{\lambda}{2}\|\boldsymbol{a}\|_2^2, \tag{8}$$

*where $f_{vv}$ is defined above.*

**Proposition 3.2** (Primal and Dual GA Characterizations). *Let $\mathcal{K}_k = J(\theta_k)J(\theta_k)^\top$. The unique solution $\boldsymbol{a}_k$ of equation 8 admits both a primal and a dual formulation:*

$$\text{(Primal)}\quad \big(J(\theta_k)^\top J(\theta_k) + \lambda I_n\big)\,\boldsymbol{a}_k = -\,J(\theta_k)^\top f_{vv}, \tag{9}$$

$$\text{(Dual)}\quad (\mathcal{K}_k + \lambda I_m)\,\boldsymbol{y}_{a,k} = -\,\mathcal{K}_k\,f_{vv}, \quad \boldsymbol{a}_k = -\tfrac{1}{\lambda}\,J(\theta_k)^\top\big(\boldsymbol{y}_{a,k} + f_{vv}\big). \tag{10}$$

**Remark 3.1** (Implementation Cost of GA). *Solving for $\boldsymbol{a}_k$ via the dual system equation 10 reuses the same operator $\mathcal{K}_k + \lambda I_m$ as the primary LM step. Hence, once a factorization or preconditioner for the dual solve is available, the only extra costs are for computing $f_{vv}$ (one Hessian–vector product or two JVPs) and solving one additional linear system with the existing operator. If using an iterative solver, the preconditioner may be shared; with a direct method, only an extra back-substitution is needed.*

### 3.3 Hessian-Free Solution of the Dual System

We aim to solve the dual linear system, previously stated in equation 6. For moderate residual dimension $m$, one can assemble $\mathcal{K}_\parallel$ and solve equation 6 with a dense Cholesky factorisation. When the total number of residual evaluations $m$ (which can be thought of as batch size in this context) is very large, however, building $\mathcal{K}_\parallel$ explicitly is not tractable. In such cases, we turn to the *Conjugate-Gradient (CG)* method (Hestenes & Stiefel, 1952; Shewchuk, 1994; Saad, 2003). CG only requires repeated applications of the linear operator $v \mapsto (\mathcal{K}_\parallel + \lambda I_m)v$ and the right-hand side vector.

**Definition 3.3** (Kernel–Vector Product). *For $v \in \mathbb{R}^m$ the action of the operator is obtained in three automatic-differentiation passes:*

1. ***Reverse mode:*** $u := J^\top v \in \mathbb{R}^n$.

2. ***Forward mode:*** $\tilde{w} := J\,u = J\,J^\top v \in \mathbb{R}^m$.

3. ***Tikhonov shift:*** $w := \tilde{w} + \lambda\,v$.

*We denote this composite map by $w = \mathrm{kvp}(v)$.*

A CG iteration therefore invokes $\mathrm{kvp}(\cdot)$ once, allowing us to solve equation 6 *Hessian-free*: neither $J$ nor $\mathcal{K}$ is ever formed explicitly, yet the exact solution is recovered through operator evaluations alone.

### 3.3.1 Low-Rank Nyström Spectral Preconditioner for the Dual System

Both the primal and residual Gramians share the same eigenvalues. In typical PINN application, they often show rapidly decaying spectra and a large spread between the top and bottom eigenvalue Ryck et al. (2024), which leads to severe ill-conditioning. To address this, we propose a spectral preconditioner. Inspired by the Nyström ideas of Frangella et al. (2023); Martinsson & Tropp (2020), we approximate the kernel $\mathcal{K}$ through a rank-truncated eigendecomposition $\mathcal{K} \approx U\hat{\Lambda}U^\top$. Here, the columns of $U \in \mathbb{R}^{m \times \ell'}$ are approximate eigenvectors and $\hat{\Lambda} \in \mathbb{R}^{\ell' \times \ell'}$ contains the corresponding approximate eigenvalues, where $\ell'$ is the effective rank determined by the Nyström procedure (typically $\ell' \le \ell$, the initial number of landmarks). We then build the operator

$$P^{-1} = U(\hat{\Lambda} + \lambda I_{\ell'})^{-1}U^\top + \frac{1}{\lambda}\big(I_m - UU^\top\big), \tag{11}$$

which *damps* the leading Nyström modes by the regularised factors $(\hat{\lambda}_i + \lambda)^{-1}$ and acts as $\lambda^{-1}I$ on the orthogonal complement. We employ $P^{-1}$ as a *left preconditioner* in Conjugate Gradient, one application of the preconditioner requires only two dense matrix–vector products with $U$ or $U^\top$. Empirically, this spectral preconditioner compresses the spread of eigenvalues and reduces CG iterations.

## 4 Algorithmic Implementation

This section outlines the implementation of the dense and iterative dual solvers. Detailed algorithms are provided in Appendix A.2.

### 4.1 Dense Dual Solver

For problems where the residual dimension $m$ is considerably smaller than the parameter dimension $n$ ($m \ll n$), a direct approach to solving the dual system (Equation equation 6) is feasible. This involves explicit formation of the regularized residual Kernel, $\widetilde{\mathcal{K}}$, and a Cholesky decomposition.

**Residual–Kernel Assembly**   Let the residual vector introduced in Sestion 2.2 be

$$r(\theta) = \big(r_\Omega(\theta),\, r_{\partial\Omega}(\theta)\big)^\top, \qquad m = N_\Omega d_\Omega + N_{\partial\Omega} d_{\partial\Omega},$$

with Jacobian split $J(\theta) = (J_\Omega;\, J_{\partial\Omega})$. The Gramian required by the dual formulations (Sections 3.1–3.3) is

$$\mathcal{K} = J\,J^\top = \begin{pmatrix} K_{\Omega\Omega} & K_{\Omega\partial\Omega} \\ K_{\partial\Omega\Omega} & K_{\partial\Omega\partial\Omega} \end{pmatrix}, \quad K_{\partial\Omega\Omega} = K_{\Omega\partial\Omega}^\top.$$

The Kernel $\mathcal{K} = JJ^\top$ is built on-the-fly, block-wise, and in parallel (Algs. 3, 2). Vector-Jacobian products (VJPs) provide Jacobian components for $\mathcal{K}$ entries; these components are then discarded, avoiding storage of the full $m \times n$ Jacobian $J$. Through Jax Bradbury et al. (2018) just-in-time compilation, The Accelerated Linear Algebra (XLA) compiler optimizes these on-the-fly computations; its operator fusion capabilities enhance speed and reduce memory overhead by preventing the explicit materialization of complete intermediate Jacobian arrays before their consumption. Moreover, symmetry is exploited for diagonal blocks. The final matrix used by the dense solver is the regularized residual Gramian $\widetilde{\mathcal{K}}_k = \mathcal{K}_k + \lambda I_m$.

**Dense Dual Solver Steps**   With the fully assembled and regularised Gramian $\widetilde{\mathcal{K}}_k$ in hand, we perform a single Cholesky factorisation and two back-substitutions to obtain both the Gauss–Newton velocity and its geodesic correction. The procedure is detailed in Algorithm 4 .

**Overall cost.**   Assembly is $\mathcal{O}(m^2n)$ (assuming JVPs/VJPs for kernel entries), Cholesky $\frac{1}{3}m^3$, and each triangular solve $\mathcal{O}(m^2)$; memory is $\mathcal{O}(m^2)$. This remains efficient for $m$ up to a few thousand.

## 4.2 Iterative Dual Solver

When the residual dimension $m$ becomes too large for the dense solver outlined in Section 4.1, forming and factorizing the $m \times m$ residual Gramian $\mathcal{K}$ (Definition 3.1) becomes computationally prohibitive. In such scenarios, we resort to an iterative method to solve the dual linear system presented in equation 6. Specifically, we employ the Preconditioned Conjugate Gradient (PCG) algorithm. For efficient PCG convergence, we employ the Nyström-based spectral preconditioner detailed in Section 3.3.1. This approach relies on the approximation $\mathcal{K} \approx U \hat{\Lambda} U^\top$ and was proposed in Arcolano (2011), the practical construction of which is outlined next. First, a small subset of $\ell \ll m$ residual components are selected as landmarks. In the context of PINNs, **a landmark corresponds to a specific scalar residual value, evaluated at a single collocation point and for a single output dimension of the residual map.** Let $I = \{i_1, \ldots, i_\ell\}$ denote the set of indices for these $\ell$ landmark residuals.

Using these landmarks, we form two key submatrices of the full Gramian $\mathcal{K}$:

- $\mathcal{K}_{II} \in \mathbb{R}^{\ell \times \ell}$: The Gramian matrix computed between the landmark residual components themselves.

- $\mathcal{K}_{CI} \in \mathbb{R}^{(m-\ell) \times \ell}$: The Gramian matrix computed between the non-landmark residual components (indexed by $C = \{1, \ldots, m\} \setminus I$) and the landmark residual components.

An eigendecomposition is performed on the (typically small) landmark Gramian: $\mathcal{K}_{II} = Q \Lambda_Q Q^\top$, where $Q \in \mathbb{R}^{\ell \times r}$ contains $r$ orthonormal eigenvectors corresponding to the $r$ positive eigenvalues in the diagonal matrix $\Lambda_Q \in \mathbb{R}^{r \times r}$ (where $r \le \ell$ is the effective rank of $\mathcal{K}_{II}$).

An extended, non-orthogonal basis $\tilde{U} \in \mathbb{R}^{m \times r}$ for the Nyström approximation is then constructed. The rows of $\tilde{U}$ corresponding to the landmark indices are set to $Q$ (i.e., $\tilde{U}_I \leftarrow Q$), while the rows corresponding to non-landmark indices are computed as $\tilde{U}_C \leftarrow \mathcal{K}_{CI} Q \Lambda_Q^{-1}$. The Nyström approximation of the Gramian is then given by $\hat{\mathcal{K}} = \tilde{U} \Lambda_Q \tilde{U}^\top$.

To obtain an orthonormal eigendecomposition $U \hat{\Lambda} U^\top$ for this approximation $\hat{\mathcal{K}}$, which is beneficial for constructing the preconditioner, we proceed as follows. Define $M = \tilde{U} \Lambda_Q^{1/2} \in \mathbb{R}^{m \times r}$. Performing a Singular Value Decomposition (SVD) on $M$ yields $M = V \Sigma W^\top$, where $V \in \mathbb{R}^{m \times r}$ has orthonormal columns, $\Sigma \in \mathbb{R}^{r \times r}$ is diagonal with singular values, and $W \in \mathbb{R}^{r \times r}$ is orthogonal. Then, the Nyström approximation can be expressed as $\hat{\mathcal{K}} = M M^\top = (V \Sigma W^\top)(V \Sigma W^\top)^\top = V \Sigma^2 V^\top$. Thus, the desired orthonormal eigenvectors are $U \leftarrow V$, and the corresponding eigenvalues are $\hat{\Lambda} \leftarrow \Sigma^2$. The complete procedure for constructing $U$ and $\hat{\Lambda}$ is detailed in Algorithm 5.

This Nyström-based spectral approximation $\hat{\mathcal{K}} \approx U \hat{\Lambda} U^\top$ is then utilized to build the preconditioner $P^{-1}$ (as defined in Equation equation 11) for the PCG algorithm. The full PCG-based solution of the dual system, incorporating this preconditioner and relying on kernel-vector products (Definition 3.3), is outlined in Algorithm 6.

## 4.3 Optimization Workflow

Given a partial differential equation (PDE) and a neural network ansatz $u_\theta$, we minimize the loss equation 1 using the described Dual Natural Gradient Descent Framework. The overall procedure, including step computation via either DenseDualSolve or PCGStep and a line search, is detailed in Algorithm 1.

---

**Algorithm 1** Dual Natural-Gradient Descent (D-NGD) Workflow

---

1: **Input:** initial parameters $\theta_0$; residual function $r_{\text{fn}}$; collocation sets $(X_\Omega, X_{\partial\Omega})$ ; loss $L(\theta)$; time budget $T_{\max}$; Levenberg–Marquardt rule for $\lambda_k$; Nyström rank $\ell$; CG tolerance $\varepsilon$; max CG its $m_{\text{CG\_max}}$; dense threshold $C_{\text{dense\_thresh}}$; flag `use_dense_GA_flag`.
2: $t_0 \leftarrow \text{now}()$
3: **while** $\text{now}() - t_0 < T_{\max}$ **do**
4: $\quad g \leftarrow \nabla_\theta L(\theta)$ {Compute $J^\top r$ at all points}
5: $\quad$ Let $\Delta\theta^\star$ be the computed parameter step.
6: $\quad m \leftarrow |X_\Omega| + |X_{\partial\Omega}|$
7: $\quad$ **if** $m < C_{\text{dense\_thresh}}$ **then**
8: $\quad\quad \Delta\theta^\star \leftarrow \text{DenseDualSolve}\big(\theta, g, r_{\text{fn}}, (X_\Omega, X_{\partial\Omega}), \lambda, \text{use\_dense\_GA\_flag}\big)$
9: $\quad$ **else**
10: $\quad\quad \Delta\theta^\star \leftarrow \text{PCGStep}\big(\theta, g, r_{\text{fn}}, \{x_s\}, \lambda, \ell, \varepsilon, m_{\text{CG\_max}}\big)$
11: $\quad$ **end if**
12: $\quad \eta \leftarrow \arg\min_{\eta\in(0,1]} L\big(\theta + \eta\,\Delta\theta^\star\big)$
13: $\quad \theta \leftarrow \theta + \eta\,\Delta\theta^\star$
14: **end while**
15:
16: **return** $\theta$

---

### 4.4 Time and Space Complexity

We summarize per-step time/memory for the primal, dense dual, and dual PCG solvers.

**Primal (parameter-space GN/LM).** The primal approach forms $G = J^\top J$ and factors an $n \times n$ system; per-step time $O(mn^2 + n^3)$ and memory $O(n^2)$. This is preferable when $m > n$ and $n$ is small enough to factor.

**Dense dual (residual-space).** The dense dual approach forms $\mathcal{K} = JJ^\top$ and factors an $m \times m$ system; per-step time $O(m^2n + m^3)$ and memory $O(m^2)$. This is preferable when $n > m$ and $m$ is small enough to factor; as a rule of thumb, if the hardware can hold $m^2$ entries at the target precision, use the dense dual Cholesky, otherwise use the iterative solver below.

**Dual PCG (Hessian-free).** The dual PCG approach avoids forming $\mathcal{K}$. Each inner iteration costs $\approx O(mn) + O(m\ell)$, so a solve with $t$ iterations costs $O\big(t(mn + m\ell)\big)$ and uses memory linear in $m, n$ plus $O(\ell^2)$. The Nyström preconditioner is built once per epoch and reused within that epoch's inner PCG solves. Total wall-time is the per-step cost times the number of outer steps $s$, and both $s$ and $t$ grow with the effective conditioning $\kappa$ (typically $t = O(\sqrt{\kappa_{\text{eff}}})$, reduced by good preconditioning).

## 5 Applications

To comprehensively assess the capabilities of Dual Natural Gradient Descent (D-NGD), we conduct a series of experiments across a diverse suite of benchmark partial differential equations. All solvers are implemented in JAX 0.5.0, we compute the derivatives using an in-house implementation of th Taylor-mode Automatic-Differentiation Bettencourt et al. (2019) unless specified.

Every run is executed on a single NVIDIA A100-80GB GPU. Unless stated otherwise, we track the *relative $L^2$ error* with respect to an analytic or DNS reference solution and report the *median* value over **ten independent** weight initializations. Instead of counting iterations, unless specified otherwise each optimizer is allocated a fixed wall-clock budget of **3000 seconds** for all experiments.

We compare the following Optimizers.

- ADAM (Kingma & Ba, 2017) with learning rate $\eta = 10^{-3}$, $(\beta_1, \beta_2) = (0.9, 0.999)$, $\epsilon = 10^{-8}$, and no weight decay.

- SGD with Nesterov momentum 0.9 and the one-cycle schedule of Smith & Topin (2018), using a peak learning rate of $10^{-2}$ and initial/final learning rates of $10^{-4}$.

- L–BFGS (Jaxopt Blondel et al. (2021) implementation) with history size 300, tolerance $10^{-6}$, maximum 20 iterations per line-search step, and a strong-Wolfe backtracking line search.

- **Dense D-NGD** — the Dense Dual–Newton natural gradient solved by a dense Cholesky factorisation (Algorithm 4), with and without geodesic acceleration (suffix "+GA").

- **PCGD-NGD** — the matrix-free CG variant (Algorithm 6), preconditioned by a Nyström approximation (Algorithm 5).

Across all benchmarks, we employ standard Multi-Layer Perceptron (MLP) architectures with tanh activation functions. This choice is motivated by our primary goal to develop an **architecture-agnostic optimization framework** and opt to use one of the most commonly used architectures in the PINN literature Wang et al. (2023). While specialized architectures can undoubtedly improve the conditioning of the loss landscape and may lead to even lower final errors Jnini et al. (2025a), our results demonstrate that a powerful, curvature-aware optimizer can achieve state-of-the-art accuracy even with a vanilla network structure.

For the dense NGD solver, we report results both *with* and *without* the second-order geodesic correction (the former denoted by the " +GA" suffix). All hyperparameter settings are listed in Appendix A. Below, we introduce our PDE benchmark and experimental setups used for the selected test cases.

## 5.1 The 10+1–Dimensional Heat Equation

We consider a heat equation in a 10-dimensional spatial domain plus time, defined for $x \in \Omega := [0,1]^{10}$ and $t \in [0,1]$, with a diffusion coefficient $\kappa = \frac{1}{4}$. The temperature field $u(t,x)$ evolves according to:

$$\partial_t u(t,x) - \kappa \Delta_x u(t,x) = 0, \qquad u(0,x) = \sum_{i=1}^{10} \sin 2\pi x_i, \qquad u|_{\partial\Omega} = 0. \tag{12}$$

The analytic solution $u_{\mathrm{ex}}(t,x) = e^{-4\pi^2 \kappa t} \sum_{i=1}^{10} \sin 2\pi x_i$ serves as ground truth. The PINN employs a tanh-MLP with layer widths $11 \to 256 \to 256 \to 128 \to 128 \to 1$, amounting to 118,401 parameters ($n \approx 118k$). Each optimiser step uses $N_\Omega = 10^4$ interior collocation points and $N_{\partial\Omega} = 10^3$ boundary and initial condition collocation points, resulting in a residual dimension $m = 6k$.

The performance of D-NGD on this problem is illustrated in Figure 1 (Left) and summarized in Table 1. As observed, Dense D-NGD+GA achieves the lowest median relative $L^2$ error of $8.52 \times 10^{-6}$, closely followed by Dense D-NGD at $1.24 \times 10^{-5}$. These results represent a substantial improvement over the best-performing baseline, L-BFGS ($9.82 \times 10^{-5}$), by more than an order of magnitude. Compared to Adam ($1.45 \times 10^{-3}$) and SGD ($3.48 \times 10^{-3}$), the D-NGD variants are over two orders of magnitude more accurate.

## 5.2 Logarithmic Fokker–Planck in $9+1$ dimensions

The Fokker–Planck equation governs the evolution of probability densities under stochastic dynamics. We map the density $p$ to its logarithm $q = \log p$ and study:

$$\partial_t q - \frac{d}{2} - \frac{1}{2} \nabla q \cdot x - \frac{1}{2} \Delta q - \|\nabla q\|^2 = 0, \qquad q(0,x) = \log p_0(x), \tag{13}$$

on $t \in [0,1]$, $x \in [-5,5]^9$. PINN formulations for this type of problem have been explored in Dangel et al. (2024); Sun et al. (2024); Hu et al. (2024); Schwencke & Furtlehner (2024). For a drift $\mu = -\frac{1}{2}x$ and diffusion $\sigma = \sqrt{2}I$, the exact density remains Gaussian, and $q^*(t,x) = \log p^*(t,x)$ is available in closed form. The PINN uses a tanh-MLP ($10 \to 256 \to 256 \to 128 \to 128 \to 1$; 118,145 parameters). The

initial condition $q(0, x) = q_0(x)$ is satisfied by construction using the transformation $u_\theta(t, x) := \psi_\theta(t, x) - \psi_\theta(0, x) + q_0(x)$, where $\psi_\theta : \mathbb{R} \times \mathbb{R}^9 \to \mathbb{R}$ is the core MLP output. Consequently, training focuses on the PDE residual, using $N_\Omega = 3000$ space-time interior points sampled per iteration. Figure 1 (Right) and Table 1 summarize the performance on the Logarithmic Fokker-Planck equation. Dense D-NGD+GA again leads with a median error of $2.47 \times 10^{-3}$, followed by Dense D-NGD at $3.27 \times 10^{-3}$. These D-NGD methods significantly outperform Adam $(4.80 \times 10^{-2})$ and SGD $(5.48 \times 10^{-2})$ by more than an order of magnitude. To the best of our knowledge, this sets a new benchmark for PINNs in forward mode for this problem.

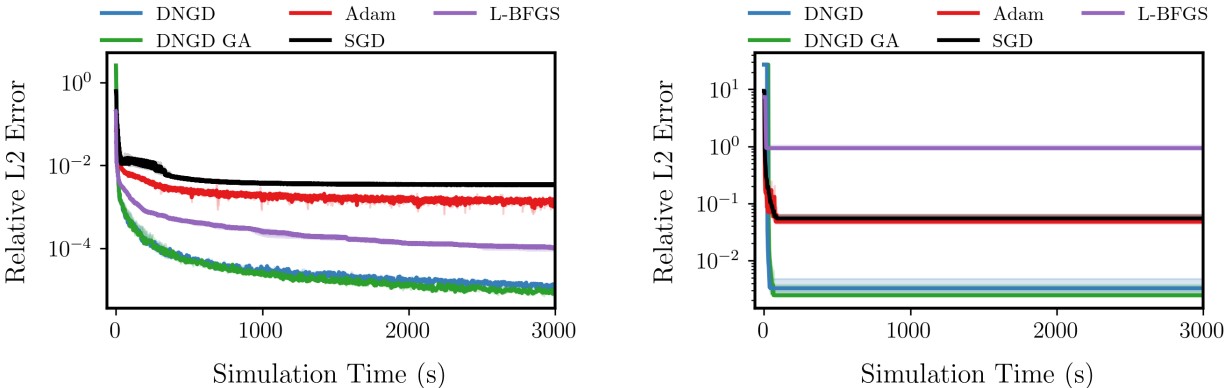

Figure 1: **Left: Heat equation in $10{+}1\,$d; Right: Logarithmic Fokker–Planck in $9{+}1\,$d.** We plot the median relative $L^2$ error across all runs, with shaded bands indicating the interquartile range (25th–75th percentiles) for all solvers.

### 5.3   Kovásznay flow at $Re = 40$

The steady two-dimensional Kovásznay solution is a classic benchmark for incompressible flow solvers Kovasznay (1948). On the domain $\Omega = [-0.5, 1.0] \times [-0.5, 1.5]$, we solve:

$$(u{\cdot}\nabla)u + \nabla p - \nu\Delta u = 0, \qquad \nabla{\cdot}\,u = 0, \qquad \nu = \tfrac{1}{40}. \tag{14}$$

Boundary conditions are prescribed from the known analytic solution. The PINN employs a tanh-MLP with four hidden layers, each with fifty neurons (7,953 parameters), similar to Jnini et al. (2024). For training, 400 interior and 400 boundary collocation points are sampled per iteration. Performance for the Kovásznay flow is depicted in Figure 2 (Left) and Table 1. Dense D-NGD achieves an exceptionally low median error of $5.23 \times 10^{-7}$, with Dense D-NGD+GA performing similarly at $5.49 \times 10^{-7}$, L-BFGS reached $9.48 \times 10^{-5}$, while Adam and SGD achieved errors of $4.50 \times 10^{-3}$ and $6.94 \times 10^{-3}$ respectively. To the best of our knowledge, this sets a new benchmark for PINNs in forward mode for this problem.

### 5.4   Allen–Cahn Reaction–Diffusion

The Allen-Cahn equation models phase separation and is a challenging benchmark due to its stiff reaction term and potential for sharp interface development, often causing difficulties for standard PINN training. We consider the equation on $(t, x) \in [0, 1] \times [-1, 1]$:

$$u_t - 10^{-4}u_{xx} + 5u^3 - 5u = 0, \tag{15}$$
$$u(0, x) = x^2 \cos \pi x,$$
$$u(t, -1) = u(t, 1), \quad u_x(t, -1) = u_x(t, 1).$$

This involves a diffusion coefficient of $10^{-4}$, a cubic reaction term, and periodic boundary conditions. The neural network is an MLP with an input layer (3 features: $t$, and $x$ after periodic embedding with period 2.0), **four hidden layers (100 neurons each)**, and one output neuron with tanh activation leading to

30,801 trainable parameters. Each training step uses $N_\Omega = 4,500$ PDE interior points and $N_{\partial\Omega} = 900$ boundary/initial condition points. Results for the Allen-Cahn equation are shown in Figure 2 (Right) and Table 1. Dense D-NGD+GA achieves the best performance with a median error of $9.13 \times 10^{-6}$, followed by Dense D-NGD at $1.21 \times 10^{-5}$. This problem particularly highlights the deficiency of first-order methods and L-BFGS, which all failed to converge to recover the physical solution in the allocated time budged. Geodesic acceleration led to a noticeably faster convergence for this problem.

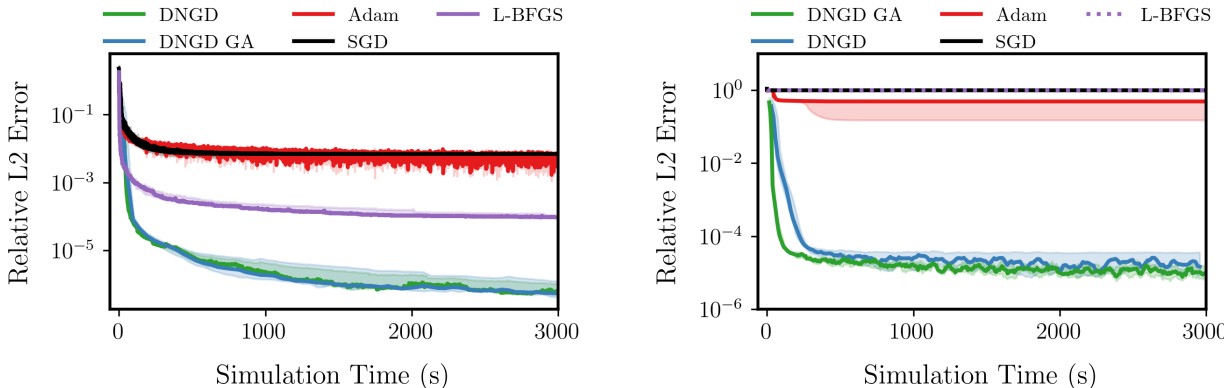

Figure 2: **Left: Kovásznay flow at $Re = 40$; Right: Allen-Cahn reaction-diffusion.** We plot the median relative $L^2$ error across all runs, with shaded bands indicating the interquartile range (25th–75th percentiles) for all solvers.

### 5.5 Lid-driven cavity at $Re = 3000$

High Reynolds number lid-driven cavity flow is a demanding benchmark where standard PINNs often struggle to achieve high accuracy beyond $Re \approx 1000$ Karniadakis et al. (2023). We consider the steady-state incompressible Navier–Stokes equations in the domain $\Omega = [0,1]^2$ with a kinematic viscosity $\nu = \frac{1}{3000}$, corresponding to $Re = 3000$. The flow is driven by a lid moving with the profile:

$$u(x,1) = 1 - \frac{\cosh[C_0(x - \frac{1}{2})]}{\cosh(\frac{1}{2}C_0)}, \qquad v(x,1) = 0, \quad C_0 = 10. \tag{16}$$

No-slip boundary conditions ($u = 0, v = 0$) are applied on the other three stationary walls. To stabilize training at this high Reynolds number, for our D-NGD method, we adopt the curriculum learning strategy described by Wang et al. (2023), involving training at progressively increasing Reynolds numbers ($Re = 100, 400,$ and $1000$) for 50 iterations each before transitioning to the target $Re = 3000$; this approach was shown to help avoid getting stuck in poor local minima. For the baseline optimizers, we employ a standard curriculum approach involving 50,000 warmup iterations. The network contains about $6.6 \times 10^4$ parameters. At each step we sample $10^4$ interior collocation points and $2 \times 10^3$ boundary points, resampling every iteration. Each run is limited to 9000 s of wall-clock time. On this problem PCGD-NGD attains a median error of $3.59 \times 10^{-3}$, over two orders of magnitude below L-BFGS at $3.85 \times 10^{-1}$ and far better than Adam at $6.93 \times 10^{-1}$ or SGD at $1.10$. which fail to accurately reconstruct the solution field. To the best of our knowledge, this sets a new benchmark for PINNs in forward mode at this Reynolds number.

### 5.6 Poisson Equation in 10 Dimensions

We consider a 10D Poisson equation, $-\Delta u(x) = f(x)$ for $x \in [0,1]^{10}$. The source $f(x)$ is from the analytical solution $u^*(x) = \sum_{k=1}^5 x_{2k-1} \cdot x_{2k}$, so the problem becomes $-\Delta u(x) = 0$ with Dirichlet boundary conditions from $u^*(x)$ on $\partial([0,1]^{10})$. The PINN is an MLP (10 inputs; **four hidden layers, 100 neurons each**; 1 output; Tanh; $\approx 41,501$ parameters). Training uses $m = 10,000$ residual points ($N_\Omega = 8,000$ interior, $N_{\partial\Omega} = 2,000$ boundary). Due to the large residual dimension $m = 10,000$, the iterative PCGD-NGD is

employed. As shown in Figure 3 (Right) and Table 1, PCGD-NGD achieves a median error of $2.74 \times 10^{-4}$. This is approximately twice as good as L-BFGS ($5.47 \times 10^{-4}$) and significantly better than Adam ($3.51 \times 10^{-2}$) and SGD ($6.26 \times 10^{-2}$) by about two orders of magnitude.

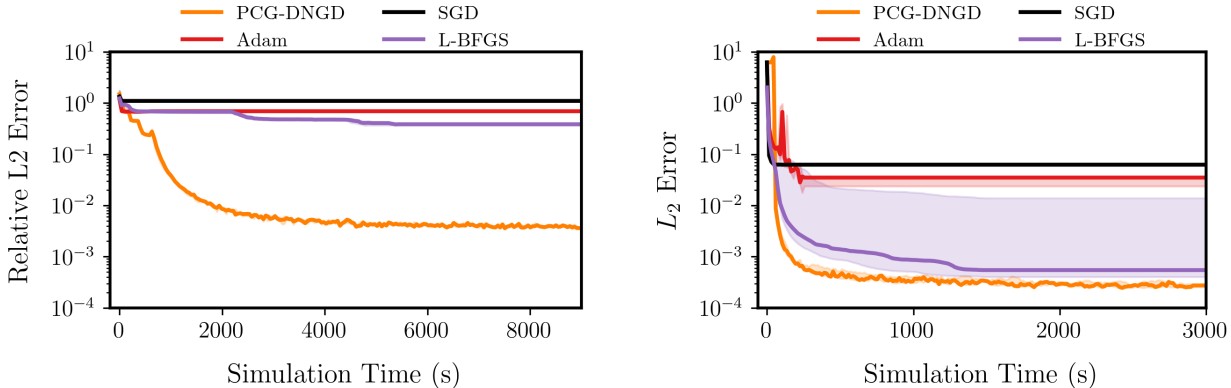

Figure 3: **Left: Lid-driven cavity at** $Re = 3000$**; Right: Poisson Equation in 10 Dimensions (with PCGD-NGD).** We plot the median relative $L^2$ error across all runs, with shaded bands indicating the interquartile range (25th–75th percentiles) for all solvers.

## 5.7 Poisson Equation in $d = 10^5$ Dimensions

To gauge the large-scale behaviour of dual–NGD we embed the inseparable two-body test function

$$u_{\text{ex}}(x) = \left(1 - \|x\|^2\right) \sum_{i=1}^{d-1} c_i\big[\sin\big(x_i + \cos x_{i+1}\big) + x_{i+1} \cos x_i\big], \qquad c_i \sim \mathcal{N}(0, 1), \tag{17}$$

into the unit ball $\mathbb{B}^{10^5}$ and pose

$$-\Delta u = f \quad \text{in } \mathbb{B}^{10^5}, \qquad u|_{\partial \mathbb{B}^{10^5}} = 0, \quad f := -\Delta u_{\text{ex}}. \tag{18}$$

A tanh-MLP with four hidden layers of 128 neurons ($\approx$ 12.8M parameters) outputs $\phi(x)$, and we define $u_\theta(x) = (1 - \|x\|^2)\,\phi(x)$, which enforces the homogeneous Dirichlet boundary by construction.

To address the prohibitive cost of evaluating every second derivative we adopt the *Stochastic Taylor Derivative Estimator* (STDE) Shi et al. (2025): for each interior collocation point $x$ a random index set $J \subset \{1, \dots, d\}$ is drawn and the Laplacian is estimated as $\frac{d}{|J|} \sum_{j \in J} \partial^2 u(x)(e_j, 0)$. Each term $\partial^2 u(x)(e_j, 0)$ is obtained in a single Taylor-mode AD pass with the 2-jet $(x, e_j, 0)$. Training proceeds with 100 Monte-Carlo interior points per step. For this $10^5$-D Poisson problem, the performance is detailed in Figure 4 and Table 1. The GA-enhanced varian tachieved a median error of $1.14 \times 10^{-5}$. This result, benefiting from an approximate 3× improvement due to geodesic acceleration over Dense D-NGD ($3.30 \times 10^{-5}$), is nearly 16× better than the best-performing baseline (Adam, $1.87 \times 10^{-4}$) and substantially lower than errors from L-BFGS and SGD. Figure 4. To the best of our knowledge, our work is the first to extend Natural-Gradient methods to PINNs of this scale.

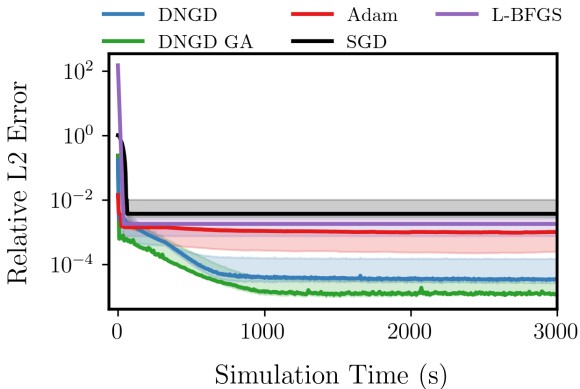

Figure 4: **Poisson Equation in** $d = 10^5$ **Dimensions** We plot the median relative $L^2$ error across all runs, with shaded bands indicating the interquartile range (25th–75th percentiles) for all solvers.

**Quantitative comparison**

Table 1 collects the final median relative $L^2$ errors for all six benchmarks.

Table 1: Median relative $L^2$ error after the task budget (ten seeds). Best performance for each experiment is highlighted in bold.

| Method | Heat (10+1d) $n{\approx}118$k $m{=}6$k | Kov. (2d) $n{\approx}8$k $m{=}0.8$k | Poisson (10d) $n{\approx}42$k $m{=}10$k | Fokker–P. (9+1d) $n{\approx}118$k $m{=}3$k | A.–Cahn (1+1d) $n{\approx}31$k $m{=}5.6$k | Cavity (2d) $n{\approx}66$k $m{=}50$k | Poisson ($10^5$d) $n{\approx}13$M $m{=}100$ |
|---|---|---|---|---|---|---|---|
| SGD | $3.48 \times 10^{-3}$ | $6.94 \times 10^{-3}$ | $6.26 \times 10^{-2}$ | $5.48 \times 10^{-2}$ | $9.93 \times 10^{-1}$ | $1.10$ | $3.67 \times 10^{-3}$ |
| Adam | $1.45 \times 10^{-3}$ | $4.50 \times 10^{-3}$ | $3.51 \times 10^{-2}$ | $4.80 \times 10^{-2}$ | $4.92 \times 10^{-1}$ | $6.93 \times 10^{-1}$ | $1.87 \times 10^{-4}$ |
| L-BFGS | $9.82 \times 10^{-5}$ | $9.48 \times 10^{-5}$ | $5.47 \times 10^{-4}$ | $9.30 \times 10^{-1}$ | $9.92 \times 10^{-1}$ | $3.85 \times 10^{-1}$ | $1.78 \times 10^{-3}$ |
| **Dense D–NGD** | $1.24 \times 10^{-5}$ | $\mathbf{5.23 \times 10^{-7}}$ | — | $3.27 \times 10^{-3}$ | $1.21 \times 10^{-5}$ | — | $3.30 \times 10^{-5}$ |
| **Dense D–NGD+GA** | $\mathbf{8.52 \times 10^{-6}}$ | $5.49 \times 10^{-7}$ | — | $\mathbf{2.47 \times 10^{-3}}$ | $\mathbf{9.13 \times 10^{-6}}$ | — | $\mathbf{1.14 \times 10^{-5}}$ |
| **PCGD-NGD** | — | — | $\mathbf{2.74 \times 10^{-4}}$ | — | — | $\mathbf{3.59 \times 10^{-3}}$ | — |

**Discussion**  Across all seven benchmarks (Table 1, Figures 1–4) the different variants of our D-NGD delivers the highest accuracy and the most reliable convergence and achieve state of the art accuracies for PINNs in several of the considered benchmarks. By swapping the intractable $n \times n$ Gauss–Newton solve in parameter space for an $m \times m$ solve in residual space, the method keeps the per-step cost proportional to the number of residuals rather than the number of weights. This single design choice lets us run many more curvature-informed iterations within the fixed budget and, equally important, frees practitioners to employ wider and deeper networks whose expressivity would otherwise be impossible to exploit. Furthermore, we have shown that geodesic acceleration (GA) provides a consistent refinement at negligible extra cost. By reusing the existing factorization and adding only one Hessian–vector product per step, GA yields 25–65% lower final errors on four of the five dense benchmarks and can speed up convergence speed without degrading performance.

## 6   Conclusion

Training high-fidelity PINNs at scale has long been hamstrung by the prohibitive cost of second-order optimisation. By revisiting the Gauss–Newton step through a primal–dual lens, **Dual Natural Gradient Descent** moves the heavy linear algebra into residual space, where it is dramatically cheaper to assemble, store, and precondition. A single Cholesky factorisation—or a handful of preconditioned CG iterations—now suffices to deliver curvature-informed updates even for networks with tens of millions of parameters. The same dual operator supports a geodesic-acceleration term at negligible extra cost, further boosting step

quality without hyper-parameter tuning. Future work will explore adaptive residual sampling driven by curvature information, automated damping strategies based on stochastic spectral estimates.

## Acknowledgments

A.J. acknowledges support from a fellowship provided by Leonardo S.p.A.

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

# A   Appendix

## A.1   Proofs of Propositions and Theorems

### A.1.1   Proof of Proposition 3.1 (Primal normal equations)

The objective function is

$$F_\lambda(\Delta\theta_k) = \tfrac{1}{2}\big(r(\theta_k) + J(\theta_k)\,\Delta\theta_k\big)^\top \big(r(\theta_k) + J(\theta_k)\,\Delta\theta_k\big) + \tfrac{\lambda}{2}\,\Delta\theta_k^\top\Delta\theta_k.$$

To find the minimizer $\Delta\theta_k^\star$, we compute the gradient of $F_\lambda(\Delta\theta_k)$ with respect to $\Delta\theta_k$ and set it to zero.
Let's expand the first term:

$$\tfrac{1}{2}\big(r(\theta_k) + J(\theta_k)\,\Delta\theta_k\big)^\top \big(r(\theta_k) + J(\theta_k)\,\Delta\theta_k\big) = \tfrac{1}{2}\big(r(\theta_k)^\top r(\theta_k) + 2r(\theta_k)^\top J(\theta_k)\,\Delta\theta_k + \Delta\theta_k^\top J(\theta_k)^\top J(\theta_k)\,\Delta\theta_k\big)$$
$$= \tfrac{1}{2}r(\theta_k)^\top r(\theta_k) + r(\theta_k)^\top J(\theta_k)\,\Delta\theta_k + \tfrac{1}{2}\Delta\theta_k^\top J(\theta_k)^\top J(\theta_k)\,\Delta\theta_k.$$

The gradient of this term with respect to $\Delta\theta_k$ is:

$$\nabla_{\Delta\theta_k}\big(\tfrac{1}{2}r(\theta_k)^\top r(\theta_k) + r(\theta_k)^\top J(\theta_k)\,\Delta\theta_k + \tfrac{1}{2}\Delta\theta_k^\top J(\theta_k)^\top J(\theta_k)\,\Delta\theta_k\big) = J(\theta_k)^\top r(\theta_k) + J(\theta_k)^\top J(\theta_k)\,\Delta\theta_k.$$

The gradient of the regularization term $\tfrac{\lambda}{2}\,\Delta\theta_k^\top\Delta\theta_k$ with respect to $\Delta\theta_k$ is:

$$\nabla_{\Delta\theta_k}\big(\tfrac{\lambda}{2}\,\Delta\theta_k^\top\Delta\theta_k\big) = \lambda\,\Delta\theta_k.$$

Combining these, the gradient of $F_\lambda(\Delta\theta_k)$ is:

$$\nabla_{\Delta\theta_k}F_\lambda(\Delta\theta_k) = J(\theta_k)^\top r(\theta_k) + J(\theta_k)^\top J(\theta_k)\,\Delta\theta_k + \lambda\,\Delta\theta_k.$$

Setting the gradient to zero for the optimal $\Delta\theta_k^\star$:

$$J(\theta_k)^\top r(\theta_k) + J(\theta_k)^\top J(\theta_k)\,\Delta\theta_k^\star + \lambda I_n\,\Delta\theta_k^\star = 0.$$

Rearranging the terms, we get:

$$\big(J(\theta_k)^\top J(\theta_k) + \lambda I_n\big)\,\Delta\theta_k^\star = -J(\theta_k)^\top r(\theta_k).$$

Given that $\nabla_\theta L(\theta_k)$ is defined as $J(\theta_k)^\top r(\theta_k)$ in this context (representing the gradient of the unweighted least-squares loss $\tfrac{1}{2}\|r(\theta_k)\|_2^2$), we have:

$$\big(J(\theta_k)^\top J(\theta_k) + \lambda I_n\big)\,\Delta\theta_k^\star = -\nabla_\theta L(\theta_k).$$

The matrix $J(\theta_k)^\top J(\theta_k)$ is positive semi-definite. For $\lambda > 0$, the matrix $J(\theta_k)^\top J(\theta_k) + \lambda I_n$ is positive definite, ensuring that $F_\lambda(\Delta\theta_k)$ is strictly convex and thus has a unique minimizer. This completes the proof of Proposition 3.1.

### A.1.2   Proof of Theorem 3.1 (Dual Normal Equations)

The Lagrangian is given by:

$$\mathscr{L}(\Delta\theta_k, y_k, \nu_k) = \tfrac{1}{2}\|r(\theta_k) + y_k\|_2^2 + \tfrac{\lambda}{2}\|\Delta\theta_k\|_2^2 + \nu_k^\top\big(y_k - J(\theta_k)\,\Delta\theta_k\big).$$

Setting the partial derivatives of $\mathscr{L}(\Delta\theta_k, y_k, \nu_k)$ with respect to $\Delta\theta_k$, $y_k$, and $\nu_k$ to zero yields the KKT system:

$$\nabla_{\Delta\theta_k}\mathscr{L}: \quad \lambda\,\Delta\theta_k - J(\theta_k)^\top\nu_k = 0, \tag{19}$$

$$\nabla_{y_k}\mathscr{L}: \quad r(\theta_k) + y_k + \nu_k = 0, \tag{20}$$

$$\nabla_{\nu_k}\mathscr{L}: \quad y_k - J(\theta_k)\,\Delta\theta_k = 0. \tag{21}$$

From equation 20, $\nu_k = -r(\theta_k) - y_k$. Substituting into equation 19 gives

$$\lambda \, \Delta\theta_k = J(\theta_k)^\top \nu_k = - \, J(\theta_k)^\top \big(r(\theta_k) + y_k\big).$$

Using $\nabla_\theta L(\theta_k) = J(\theta_k)^\top r(\theta_k)$ (gradient of the unweighted least-squares loss $\frac{1}{2}\|r(\theta_k)\|_2^2$), this becomes $\lambda \, \Delta\theta_k = -\nabla_\theta L(\theta_k) - J(\theta_k)^\top y_k$, which rearranges to Equation equation 7:

$$\Delta\theta_k^\star = -\frac{1}{\lambda}\big(J(\theta_k)^\top y_k^\star + \nabla_\theta L(\theta_k)\big).$$

Substituting this expression for $\Delta\theta_k^\star$ into equation 21 yields for the optimal $y_k^\star$:

$$y_k^\star = J(\theta_k)\,\Delta\theta_k^\star = -\tfrac{1}{\lambda}\,J(\theta_k)\,\big(J(\theta_k)^\top y_k^\star + \nabla_\theta L(\theta_k)\big).$$

Multiplying by $\lambda$ and expanding gives:

$$\lambda y_k^\star = -J(\theta_k)J(\theta_k)^\top y_k^\star - J(\theta_k)\nabla_\theta L(\theta_k).$$

Using the definition $\mathcal{K}_k = J(\theta_k)J(\theta_k)^\top$ (Definition 3.1), we have:

$$\lambda y_k^\star = -\mathcal{K}_k y_k^\star - J(\theta_k)\nabla_\theta L(\theta_k).$$

Rearranging gives $(\mathcal{K}_k + \lambda I_m)y_k^\star = -J(\theta_k)\nabla_\theta L(\theta_k)$, which is Equation equation 6. The equivalence between the primal system equation 3.1 and this dual formulation can be confirmed by substitution. Because the objective is strictly convex and the equality constraints are affine with a non-empty feasible set, the KKT conditions are necessary and sufficient and strong duality holds. This completes the proof of Theorem 3.1.

### A.1.3 Proof of Proposition 3.2 (Primal and Dual GA Characterizations)

First, the *primal* condition follows by differentiating $\frac{1}{2}\|J(\theta_k)\boldsymbol{a} + f_{vv}\|^2 + \frac{\lambda}{2}\|\boldsymbol{a}\|^2$ w.r.t. $\boldsymbol{a}$:

$$\nabla_{\boldsymbol{a}}\Big(\tfrac{1}{2}\|J(\theta_k)\boldsymbol{a} + f_{vv}\|^2 + \tfrac{\lambda}{2}\|\boldsymbol{a}\|^2\Big) = J(\theta_k)^\top\big(J(\theta_k)\boldsymbol{a} + f_{vv}\big) + \lambda\,\boldsymbol{a},$$

and setting this to zero yields equation 9:

$$\big(J(\theta_k)^\top J(\theta_k) + \lambda I_n\big)\,\boldsymbol{a}_k = -\,J(\theta_k)^\top f_{vv}.$$

To derive the *dual* formulation, we introduce the auxiliary variable $\boldsymbol{y} = J(\theta_k)\boldsymbol{a}$ and enforce the constraint $\boldsymbol{y} = J(\theta_k)\boldsymbol{a}$ via Lagrange multiplier $\boldsymbol{\nu} \in \mathbb{R}^m$. The Lagrangian for the minimization problem in equation 8 is

$$\mathcal{L}(\boldsymbol{a}, \boldsymbol{y}, \boldsymbol{\nu}) = \tfrac{1}{2}\|\boldsymbol{y} + f_{vv}\|_2^2 + \tfrac{\lambda}{2}\|\boldsymbol{a}\|_2^2 + \boldsymbol{\nu}^\top(\boldsymbol{y} - J(\theta_k)\boldsymbol{a}).$$

The KKT conditions are obtained by setting the partial derivatives to zero:

$$\nabla_{\boldsymbol{a}}\mathcal{L} : \lambda\,\boldsymbol{a} \;-\; J(\theta_k)^\top \boldsymbol{\nu} = 0,$$
$$\nabla_{\boldsymbol{y}}\mathcal{L} : \boldsymbol{y} + f_{vv} + \boldsymbol{\nu} = 0,$$
$$\nabla_{\boldsymbol{\nu}}\mathcal{L} : \boldsymbol{y} - J(\theta_k)\boldsymbol{a} = 0.$$

From $\nabla_{\boldsymbol{y}}\mathcal{L} = 0$, we get $\boldsymbol{\nu} = -(\boldsymbol{y} + f_{vv})$. Substituting this into $\nabla_{\boldsymbol{a}}\mathcal{L} = 0$ gives $\lambda\,\boldsymbol{a} = -J(\theta_k)^\top\big(\boldsymbol{y} + f_{vv}\big)$, which, for the optimal $\boldsymbol{a}_k$ and corresponding $\boldsymbol{y}_{a,k}$, matches the expression for $\boldsymbol{a}_k$ in equation 10:

$$\boldsymbol{a}_k = -\tfrac{1}{\lambda}\,J(\theta_k)^\top\big(\boldsymbol{y}_{a,k} + f_{vv}\big).$$

Meanwhile, from $\nabla_{\boldsymbol{\nu}}\mathcal{L} = 0$, we have $\boldsymbol{y} = J(\theta_k)\boldsymbol{a}$. Substituting the expression for $\boldsymbol{a}$:

$$\boldsymbol{y} = J(\theta_k)\Big(-\tfrac{1}{\lambda}\,J(\theta_k)^\top\big(\boldsymbol{y} + f_{vv}\big)\Big) = -\tfrac{1}{\lambda}\big(J(\theta_k)J(\theta_k)^\top\big)\,(\boldsymbol{y} + f_{vv}).$$

Using $\mathcal{K}_k = J(\theta_k)J(\theta_k)^\top$, this becomes

$$\boldsymbol{y} = -\tfrac{1}{\lambda}\,\mathcal{K}_k\,(\boldsymbol{y} + f_{vv}).$$

Rearranging gives $\lambda\boldsymbol{y} = -\mathcal{K}_k\boldsymbol{y} - \mathcal{K}_k f_{vv}$, so $(\mathcal{K}_k + \lambda I_m)\,\boldsymbol{y} = -\mathcal{K}_k\,f_{vv}$. For the optimal $\boldsymbol{y}_{a,k}$, this is the first part of equation 10. This completes the proof of Proposition 3.2.

## A.2 Algorithmic Implementations

This section provides the detailed algorithms referenced in the main text.

---

**Algorithm 2** KernelEntry via double VJP

---

1: **Input:** collocation points $x_i, x_j$, parameters $\theta$, residual fn. $r_{\text{fn}}$
2: Define $f_i(\theta) = r_{\text{fn}}(x_i, \theta), \ f_j(\theta) = r_{\text{fn}}(x_j, \theta)$
3: $(y_j, \ \text{vjp}_j) \leftarrow \text{jax.vjp}(f_j, \theta)$
4: $(y_i, \ \text{vjp}_i) \leftarrow \text{jax.vjp}(f_i, \theta)$
5: Initialize $B \leftarrow 0_{d \times d}$ {$d$ = dim of each residual}
6: **for** $k = 1$ **to** $d$ **do**
7:    $u \leftarrow \text{vjp}_j(e_k)$ {backprop seed $e_k$ through $f_j$}
8:    $v \leftarrow \text{vjp}_i(u)$ {backprop $u$ through $f_i$}
9:    $B_{:,k} \leftarrow v$ {column $k$ of $B$}
10: **end for**
11: **return** $B$ $\{J_i J_j^\top\}$

---

### A.2.1 Algorithm: Assemble Residual Gramian $\widetilde{\mathcal{K}}$

---

**Algorithm 3** Assemble residual Gramian $\mathcal{K}$

---

1: **Input:** collocation points $\{x_s\}_{s=1}^m$, parameters $\theta$, residual fn. $r_{\text{fn}}$
2: $n_1 \leftarrow N_\Omega d_\Omega, \quad n_2 \leftarrow N_{\partial\Omega} d_{\partial\Omega}$ {type split}
3: $\mathcal{K} \leftarrow 0_{m \times m}$
4: **for** $i \leftarrow 1$ **to** $m$ **do**
5:   **for** $j \leftarrow 1$ **to** $m$ **do**
6:     **if** $(i \le n_1 \wedge j \le n_1 \wedge j < i) \ \vee \ (i > n_1 \wedge j > n_1 \wedge j < i)$ **then**
7:       **continue**
8:     **end if**
9:     $B \leftarrow \textsc{KernelEntry}(x_i, x_j, \theta, r_{\text{fn}})$
10:     $\mathcal{K}_{ij} \leftarrow B$
11:     **if** $i \ne j$ **or** $(i \le n_1 \wedge j > n_1)$ **or** $(i > n_1 \wedge j \le n_1)$ **then**
12:       $\mathcal{K}_{ji} \leftarrow B^\top$
13:     **end if**
14:   **end for**
15: **end for**
16: **return** $\mathcal{K}$

---

### A.2.2 Algorithm:Dense Dual Solve

---

**Algorithm 4** Dense dual solver step (DENSEDUALSOLVE)

---

1: **Input:** $\theta$, $g_\theta = \nabla_\theta L(\theta)$, $r_{\mathrm{fn}}$, collocation sets $(X_\Omega, X_{\partial\Omega})$, damping $\lambda$, flag `use_geodesic_acceleration`
2: $\mathcal{K} \leftarrow \mathrm{ASSEMBLEGRAMIAN}(\text{all points}, \theta, r_{\mathrm{fn}})$
3: $\widetilde{\mathcal{K}} \leftarrow \mathcal{K} + \lambda I_m$
4: $L_{\mathrm{chol}} \leftarrow \mathrm{chol}(\widetilde{\mathcal{K}})$
5: $b \leftarrow -J(\theta)\, g_\theta$
6: Solve $L_{\mathrm{chol}}\, y = b$ then $L_{\mathrm{chol}}^\top y^\star = y$
7: $v \leftarrow -\lambda^{-1}\big(J(\theta)^\top y^\star + g_\theta\big)$
8: **if** `use_geodesic_acceleration` **then**
9: $\quad f_{vv} \leftarrow \frac{d^2}{dt^2}\, r_{\mathrm{fn}}(\text{points}, \theta + tv)\big|_{t=0}$
10: $\quad b_a \leftarrow -J(\theta)J(\theta)^\top f_{vv}$
11: $\quad$ Solve $L_{\mathrm{chol}}\, y_a = b_a$ then $L_{\mathrm{chol}}^\top y_a^\star = y_a$
12: $\quad a \leftarrow -\lambda^{-1}\big(J(\theta)^\top (y_a^\star + f_{vv})\big)$
13: $\quad \Delta\theta \leftarrow v$
14: $\quad$ **if** $\|v\|_2 > \epsilon_{\mathrm{norm}}$ **then**
15: $\qquad r \leftarrow 2\,\|a\|_2 / \|v\|_2$
16: $\qquad$ **if** $r \leq 0.5$ **then**
17: $\qquad\quad \Delta\theta \leftarrow \Delta\theta + 0.5\, a$
18: $\qquad$ **end if**
19: $\quad$ **end if**
20: **else**
21: $\quad \Delta\theta \leftarrow v$
22: **end if**
23: **return** $\Delta\theta$

---

### A.3 Algorithm:Preconditioned Conjugate Gradient

---

**Algorithm 5** Nyström construction of $U, \hat{\Lambda}$

---

1: **Input:** $\theta$, residual pts. $\{x_s\}_1^m$, $r_{\mathrm{fn}}$, landmarks $\ell$
2: Choose landmark index set $I$, $|I| = \ell$ ; let $C = \{1, \ldots, m\} \setminus I$
3: Form $\mathcal{K}_{II}$ and $\mathcal{K}_{CI}$ via KERNELENTRY
4: Eigendecompose $\mathcal{K}_{II} = Q\Lambda_Q Q^\top$ {$r$ positive eigenpairs}
5: $\tilde{U}_I \leftarrow Q$ ; $\tilde{U}_C \leftarrow \mathcal{K}_{CI}Q\Lambda_Q^{-1}$
6: SVD $\tilde{U}\Lambda_Q^{1/2} = V\Sigma W^\top$
7: $U \leftarrow V$ ; $\hat{\Lambda} \leftarrow \Sigma^2$
8: **return** $(U, \hat{\Lambda})$

---

---

**Algorithm 6** Pre-conditioned CG step (PCGSTEP)
1: **Input:** $\theta$, $g = \nabla_\theta L(\theta)$, $r_{\text{fn}}$, $\{x_s\}$, $\lambda$, rank $\ell$, tol. $\varepsilon$, $m_{\max}$
2: $(U, \hat{\Lambda}) \leftarrow$ BUILDNYSTROMAPPROXIMATION$(\theta, \{x_s\}, r_{\text{fn}}, \ell)$
3: Build preconditioner $P^{-1}$ using $(U, \hat{\Lambda}, \lambda)$
4: $b \leftarrow -J(\theta)g$ ; $y \leftarrow 0$ ; $r \leftarrow b$
5: $z \leftarrow P^{-1}r$ ; $p \leftarrow z$ ; $\rho \leftarrow \langle r, z \rangle$
6: **for** $j \leftarrow 0$ **to** $m_{\max} - 1$ **do**
7:    $Ap \leftarrow \mathcal{K}p + \lambda p$ {two JVP/VJP calls}
8:    **if** $\langle p, Ap \rangle \approx 0$ **then**
9:       **break**
10:    **end if**
11:    $\alpha \leftarrow \rho / \langle p, Ap \rangle$
12:    $y \leftarrow y + \alpha p$ ; $r \leftarrow r - \alpha Ap$
13:    **if** $\|r\|_2 \leq \varepsilon \|b\|_2$ **then**
14:       **break**
15:    **end if**
16:    $z \leftarrow P^{-1}r$ ; $\rho_{\text{new}} \leftarrow \langle r, z \rangle$
17:    **if** $\rho \approx 0$ **then**
18:       **break**
19:    **end if**
20:    $\beta \leftarrow \rho_{\text{new}} / \rho$ ; $p \leftarrow z + \beta p$ ; $\rho \leftarrow \rho_{\text{new}}$
21: **end for**
22: $\Delta\theta \leftarrow -\lambda^{-1}\left(J(\theta)^\top y + g\right)$
23: **return** $\Delta\theta$

---

### A.3.1 Algorithm: Dual Natural-Gradient Descent Workflow

### A.4 Hyperparameters for Allen–Cahn Reaction–Diffusion Experiment

Table 2: 1 + 1-D Allen–Cahn, $(t, x) \in [0, 1] \times [-1, 1]$, diffusion $10^{-4}$.

| Category | Setting |
|---|---|
| PDE | $u_t - 10^{-4}u_{xx} + 5u^3 - 5u = 0$; periodic BCs; $u(0, x) = x^2 \cos(\pi x)$ |
| Network | MLP, tanh; layers $[3, 100, 100, 100, 100, 1]$; $\approx 30\,801$ params |
| Training | $N_\Omega = 4\,500$, $N_{\partial\Omega} = 900$; budget 3 000 s; 10 seeds |
| Dense D-NGD | Dense Cholesky; $\lambda_k = \min(\text{loss}, 10^{-5})$; 31-pt line search; with/without GA |
| Adam | $\eta = 10^{-3}$; $(\beta_1, \beta_2) = (0.9, 0.999)$; $\epsilon = 10^{-8}$ |
| SGD | One-cycle (peak $5 \times 10^{-3}$, final $10^{-4}$); momentum 0.9 |
| L-BFGS | Jaxopt; history 300; strong-Wolfe; tol $10^{-6}$ |

### A.5 Hyperparameters for 10+1-Dimensional Heat Equation Experiment

Table 3: Heat on $[0, 1]^{10} \times [0, 1]$, $\kappa = \frac{1}{4}$.

| | |
|---|---|
| Network | MLP, tanh; $[11, 256, 256, 128, 128, 1]$; 118 401 params |
| Training | $N_\Omega = 10\,000$, $N_{\partial\Omega} = 1\,000$; budget 3 000 s; 10 seeds |
| Dense–DNGD | $\lambda_k = \min(\text{loss}, 10^{-3})$; 31-pt line search; GA variant identical |
| Baselines | Adam / SGD / L-BFGS as in Table 2 |

## A.6  Hyperparameters for Logarithmic Fokker–Planck Experiment

Table 4: Eq. equation 13 on $x \in [-5, 5]^9$, $t \in [0, 1]$.

| Network | MLP, tanh; $[10, 256, 256, 128, 128, 1]$; $118\,145$ params |
|---|---|
| Training | Interior residuals only, $N_\Omega = 3\,000$; budget 3 000 s; 10 seeds |
| Dense–DNGD | $\lambda_k = \min(\text{loss}, 10^{-5})$; 31-pt line search; GA variant identical |
| Baselines | Adam / SGD / L-BFGS as in Table 2 |

## A.7  Hyperparameters for Kovásznay Flow Experiment

Table 5: Steady 2-D Kovásznay benchmark ($Re = 40$).

| Network | MLP, tanh; $[2, 50, 50, 50, 50, 3]$; $7\,953$ params |
|---|---|
| Training | $N_\Omega = 400$, $N_{\partial\Omega} = 400$; budget 3 000 s; 10 seeds |
| Dense–DNGD | $\lambda_k = \min(\text{loss}, 10^{-5})$; 31-pt line search; GA variant identical |
| Baselines | Adam / SGD / L-BFGS as in Table 2 |

## A.8  Hyperparameters for Lid-Driven Cavity Experiment

Table 6: Steady lid-driven cavity ($Re = 3000$; budget 9 000 s).

| Network | MLP, tanh; $[2, 128, 128, 128, 128, 128, 3]$; $\approx 66\,000$ params |
|---|---|
| Training | $N_\Omega = 10\,000$, $N_{\partial\Omega} = 2\,000$; 10 seeds |
| Curriculum (DNGD) | $Re = 100, 400, 1000$ (50 it. each) $\rightarrow Re = 3000$ |
| Curriculum (baselines) | 50 000 warm-up iterations at each $Re = 100, 400, 1000$ (per Wang et al. (2023)) $\rightarrow Re = 3000$ |
| Iterative PCGD-NGD | Nyström rank 2500; CG tol $10^{-10}$; max 500; $\lambda_k = \min(\text{loss}, 10^{-5})$; 31-pt line search |
| Baselines | Adam (exp-decay LR), SGD (one-cycle), L-BFGS (history 300) |

## A.9  Hyperparameters for 10-Dimensional Poisson Experiment

Table 7: Laplace on $[0, 1]^{10}$ with analytic Dirichlet BCs.

| Network | MLP, tanh; $[10, 100, 100, 100, 100, 1]$; $41\,501$ params |
|---|---|
| Training | $N_\Omega = 8\,000$, $N_{\partial\Omega} = 2\,000$; budget 3 000 s; 10 seeds |
| Iterative PCGD-NGD | Nyström rank 2500; CG tol $10^{-10}$; max 500; $\lambda_k = \min(\text{loss}, 10^{-5})$; 31-pt line search; *no GA* |
| Baselines | Adam / SGD / L-BFGS as in Table 2 |

## A.10  Hyperparameters for $10^5$-Dimensional Poisson Experiment

Table 8: Poisson on $\mathbb{B}^{10^5}$

| Network | MLP, tanh; $[100000, 128, 128, 128, 128, 1]$; $\approx 12.8\,\text{M}$ params |
|---|---|
| Training | $N_\Omega = 100$ (STDE, re-sample each step); budget 3 000 s; 10 seeds |
| Dense–DNGD | Dense Cholesky; $\lambda_k = \min(\text{loss}, 10^3)$; 5-pt line search |
| Dense–DNGD +GA | Same with geodesic acceleration |
| Baselines | Adam / SGD / L-BFGS as in Table 2 |

### A.11 Navier–Stokes $(m \times n)$ Sweep: Primal GN vs. Dual D-NGD (500 iters)

We report the wall-time required to complete 500 optimizer iterations for the steady Navier–Stokes setup across a grid of residual counts $m$ and parameter counts $n$. Two solvers are compared: (i) *Primal* Gauss–Newton/Levenberg–Marquardt in parameter space; (ii) *Dual* D-NGD, which computes the same step in residual space. All experiments in this section were run on an **NVIDIA A100** GPU.

**Experimental note.** Each cell in the following two tables shows the average time per iteration in seconds, calculated from a total run of 500 iterations for the given $(m, n)$.

Table 9: Primal (parameter-space) Gauss–Newton: time per iteration (seconds) across $(m, n)$.

| $m$ | $n = 303$ | $n = 2853$ | $n = 5403$ | $n = 7953$ | $n = 10503$ | $n = 13053$ | $n = 15603$ | $n = 18153$ |
|---|---|---|---|---|---|---|---|---|
| $m = 300$ | 0.0012 | 0.0054 | 0.0146 | 0.0294 | 0.0480 | 0.0824 | 0.1226 | 0.1681 |
| $m = 1800$ | 0.0039 | 0.0210 | 0.0419 | 0.0676 | 0.1062 | 0.1554 | 0.2138 | 0.2868 |
| $m = 3300$ | 0.0045 | 0.0556 | 0.0969 | 0.1496 | 0.2121 | 0.2898 | 0.3857 | 0.4917 |
| $m = 4800$ | 0.0060 | 0.1031 | 0.1805 | 0.2705 | 0.3734 | 0.4913 | 0.6278 | 0.7791 |
| $m = 6300$ | 0.0084 | 0.1732 | 0.2985 | 0.4389 | 0.5950 | 0.7675 | 0.9606 | — |

Table 10: Dual D-NGD: time per iteration (seconds).

| $m$ | $n = 303$ | $n = 2853$ | $n = 5403$ | $n = 7953$ | $n = 10503$ | $n = 13053$ | $n = 15603$ | $n = 18153$ |
|---|---|---|---|---|---|---|---|---|
| $m = 300$ | 0.0015 | 0.0025 | 0.0035 | 0.0043 | 0.0053 | 0.0061 | 0.0069 | 0.0078 |
| $m = 1800$ | 0.0059 | 0.0072 | 0.0098 | 0.0127 | 0.0138 | 0.0164 | 0.0188 | 0.0213 |
| $m = 3300$ | 0.0069 | 0.0120 | 0.0179 | 0.0232 | 0.0266 | 0.0307 | 0.0325 | 0.0418 |
| $m = 4800$ | 0.0099 | 0.0194 | 0.0290 | 0.0344 | 0.0440 | 0.0466 | 0.0511 | 0.0657 |
| $m = 6300$ | 0.0161 | 0.0308 | 0.0416 | 0.0591 | 0.0639 | 0.0726 | 0.0807 | — |

Table 11: Primal vs. Dual regime map (empirical): winner at each $(m, n)$ based on measured time per iteration. Green = Dual faster; Orange = Primal faster.

| $m$ | $n = 303$ | $n = 2853$ | $n = 5403$ | $n = 7953$ | $n = 10503$ | $n = 13053$ | $n = 15603$ | $n = 18153$ |
|---|---|---|---|---|---|---|---|---|
| $m = 300$ | Primal | Dual | Dual | Dual | Dual | Dual | Dual | Dual |
| $m = 1800$ | Primal | Dual | Dual | Dual | Dual | Dual | Dual | Dual |
| $m = 3300$ | Primal | Dual | Dual | Dual | Dual | Dual | Dual | Dual |
| $m = 4800$ | Primal | Dual | Dual | Dual | Dual | Dual | Dual | Dual |
| $m = 6300$ | Primal | Dual | Dual | Dual | Dual | Dual | Dual | — |

### A.12 Effect of the Number of Landmarks on the Nyström Preconditioner

The efficiency of the Preconditioned Conjugate Gradient (PCG) solver hinges on the quality of the preconditioner. Our choice of a Nyström-based spectral preconditioner is motivated by the empirical observation that the PINN Gramian matrix, $\mathcal{K}$, often exhibits a rapidly decaying eigenspectrum. This property implies that a low-rank approximation can effectively capture the dominant spectral information responsible for ill-conditioning.

To substantiate this, we conducted an analysis on the Lid-driven cavity benchmark at Re=3000, where the number of residuals is large ($m = 50,000$). Figure 5 plots the eigenvalues of the Gramian matrix at different stages of training, clearly illustrating the rapid spectral decay where a small fraction of eigenvalues contains most of the spectral energy. The analysis in Table 12 further demonstrates the practical benefit of this property, showing a sharp decrease in the number of PCG iterations required for convergence as the

number of landmarks ($l$) in the Nyström approximation increases. A relatively small $l \ll m$ is sufficient to significantly accelerate the solver, confirming the effectiveness and efficiency of our preconditioning strategy.

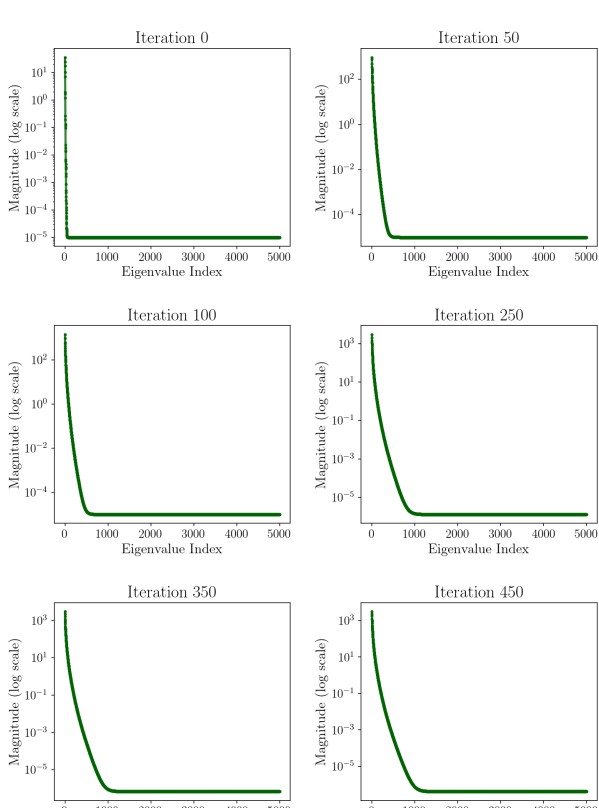

Figure 5: Evolution of the first 5000 eigenvalues of the Gramian matrix at different training iterations for the Lid-driven cavity problem. The plots show that the spectrum decays rapidly in the earlier stages of training.

Table 12: Convergence of the Preconditioned Conjugate Gradient (PCG) solver as a function of training iteration and the number of landmarks ($l$) used in the Nyström approximation. Each cell shows the number of iterations required to reach the convergence tolerance. A value of 500 indicates that the solver did not converge within the maximum allowed iterations, highlighting the necessity of a sufficiently large rank for the preconditioner, particularly in later, more challenging stages of training.

| Epoch | CG Iterations for Number of Landmarks ($l$) | | | | |
|---|---|---|---|---|---|
| | 500 | 1000 | 2500 | 4000 | 5000 |
| 0 | 10 | 10 | 10 | 10 | 10 |
| 50 | 500 | 172 | 11 | 10 | 10 |
| 100 | 500 | 337 | 20 | 11 | 11 |
| 250 | 500 | 500 | 230 | 38 | 17 |
| 350 | 500 | 500 | 338 | 60 | 23 |
| 450 | 500 | 500 | 457 | 80 | 31 |

