# OpenReview forum: "Dual Natural Gradient Descent for Scalable Training of Physics-Informed Neural Networks"
_TMLR — Accepted by TMLR_

### Review · Reviewer_Rjch · 2025-06-23

**Summary Of Contributions:**

Physics-Informed Neural Networks (PINNs) are proposed as a neural approach to solving partial differential equations more efficiently. However, they often suffer from training instabilities, which has been hypothesized to be due to difficult optimizations. Natural Gradient Descent (NGD) has been proposed to improve optimizations, yet previous attempts often do not scale to large models. The paper proposes to perform NGD in a dual space given by the residuals, thus improving the scalabilities. Using geodesic acceleration as a correction term was shown to improve the performances. Nyström spectral preconditioner helps to improve the computation speed for large batch sizes. The benefits of the proposed approach are demonstrated on several PINN benchmarks, showing improvements upon the baselines.

**Audience:**

Yes

**Broader Impact Concerns:**

The work is largely theoretical, and I do not find major concerns.

**Claims And Evidence:**

Yes

**Requested Changes:**

I am not an expert in PINN, but I am reasonably familiar with natural gradient descent. I find the explorations on improving the training of PINNs using NGD interesting, while not being able to determine their significance due to lack of familarities. However, it would be beneficial if the authors can discuss the relation of the proposed approaches to the following works:

1. While not its core contribution, similar idea on efficiently solving for natural gradient update was presented in Appendix I of Benzing 2022, where it was demonstrated that the exact NGD update can be obtained in a lower dimensional space by invoking the matrix inversion lemma. This appears related to the proposed primal-dual view.

2. Song et al. 2018 proposed to accelerate NGD by accounting for the  geodesic acceleration. This appears rather related to the geodesic acceleration method proposed in the paper.

Benzing, Gradient Descent on Neurons and its Link to Approximate Second-Order Optimization, ICML 2022

Song et al., Accelerating Natural Gradient with Higher-Order Invariance, ICML 2018

**Strengths And Weaknesses:**

Strengths:

1. The proposed approaches are theoretically principled,
2. Empirical evidence demonstrates improvements upon the chosen baselines while taking the uncertainties into consideration.

Weaknesses:

1. Some potential related works are not discussed. See Requested Changes.
2. Relatedly, Song et al. 2018. noted that using the geodesic acceleration is not an essential component of NGD itself, but rather an approximation of the Riemannian Euler method. It might be beneficial to discuss this perspective further in the paper.

Song et al., Accelerating Natural Gradient with Higher-Order Invariance, ICML 2018

---

> ### Author Response · Authors · 2025-08-12
>
> We thank the reviewer for the insightful comments and references, which help us clarify the contributions and context of our work.
>
>
>
> ### On the Relation to Benzing et al. (2022)
>
> We thank the reviewer for bringing this work to our attention. The efficient update in our work is indeed related to the technique in Appendix I of Benzing et al., which applies a related method to the Fisher Information Matrix and which is a consequence of the Pushforward Identity[1]. From the convex optimization perspective we adopt, we arrive at a similarly efficient formulation via a different route: a first-principles derivation using Karush–Kuhn–Tucker  conditions on the regularized least-squares problem. We believe this connection is important and, more broadly, links the primal–dual formalism to the Pushforward Identity and the “kernel trick” widely used in kernel methods but still underexplored in the Natural Gradient and PINN communities. We have revised the manuscript in consequence to include this discussion in the related work section, with changes highlighted in blue.
>
>
> ---
>
> ### On the Relation to Song et al. (2018) and Geodesic Acceleration
>
> The reviewer is correct that the geodesic correction we employ is related to the method of Song et al. and can be interpreted as an approximation of the Riemannian Euler method. Our primary contribution here is to show that this acceleration term can be computed with negligible overhead within our dual framework, making it practical and scalable. We added a discussion of this connection to the related work section.
>
>
> We hope to have addressed the reviewer's questions and remain at their disposal should further clarifications be needed.
>
>
> References:
>
> [1] H. V. Henderson and S. R. Searle. On deriving the inverse of a sum of matrices. SIAM Review, 23(1):53–60,
> 1981.

---

> > ### Comment · Reviewer_Rjch · 2025-08-18
> >
> > I thank the authors for their response, which addressed my concerns.

---

### Review · Reviewer_U1wM · 2025-07-07

**Summary Of Contributions:**

The submission addresses the challenges faced in solving Physics-Informed Neural Networks (PINNs) by comparing various optimization methods. It highlights that traditional approaches, such as Adam and SGD, do not yield satisfactory accuracy, making Gauss-Newton a more favorable choice due to its ability to utilize geometric information from the functional space. However, Gauss-Newton's time and memory complexities remain significant, particularly concerning the parameter and residual space sizes.

To overcome these limitations, the submission introduces the Dual Formulation via KKT in the residual space, which allows for a reduced time complexity compared to the conventional Gauss-Newton method. Additionally, it integrates geodesic acceleration to enhance accuracy further. Recognizing the computational difficulties associated with the Hessian matrix, the authors propose a Hessian-free solution for the dual system, employing the Conjugate Gradient method for efficient approximation. This innovative approach provides a pathway to improve the performance of PINNs while managing computational resources effectively.

**Audience:**

Yes

**Broader Impact Concerns:**

The work presents no significant ethical concerns.

**Claims And Evidence:**

Yes

**Requested Changes:**

See the above questions and weaknesses.

**Strengths And Weaknesses:**

The submission demonstrates several strong aspects, particularly in its mathematical rigor and clarity. It presents well-defined algorithms and proofs, showcasing a robust theoretical foundation. The introduction of the Dual Natural Gradient PINN is noteworthy, supported by extensive experimental results across various scenarios that effectively illustrate the generalizability of the research. Additionally, the paper integrates a diverse range of ideas into the context of PINNs, paving the way for potential future research directions.

However, there are some weaknesses that require attention. While the paper emphasizes the advantages of DNGD as an improved version of Gauss-Newton, it primarily compares this method with first-order techniques like Adam and SGD. A comparison with other second-order methods is essential, especially given the claim of superior time complexity. It would be beneficial to evaluate how quickly and effectively DNGD converges to a low Relative L2 error relative to these alternatives.

Furthermore, in the Iterative Dual Solver process, the mention of the number of landmarks for the Nyström approximation is vague, only stating that it is less than a certain value. A more detailed explanation of what constitutes an appropriate value for this parameter is needed, along with specifics on its usage in the experiments. This applies to the variable in the Low-Rank Nyström Spectral Preconditioner for the Dual System as well. Without clear information on these values, it is challenging to support claims of effective approximation.

Lastly, the variable Cdense_thresh in the algorithm section lacks context. Given the paper's focus on reducing time complexity, criteria for this variable are necessary to assess whether the claimed advantages in time complexity are valid. Providing such details would strengthen the overall submission and enhance its credibility.


Here is the additional questions:

### Questions

- How does D-NGD compare to the Gauss-Newton method in terms of results?

- While I believe that time complexity alone is sufficient since it is higher than memory complexity, could the paper also address the advantage in memory complexity as a metric?

- The degree of approximation of \ell in the Low-Rank Nyström Spectral Preconditioner for the Dual System likely varies significantly depending on the value of the variable. Could you explain how \ell is determined or how results vary based on this value?

- The algorithm applies different methods based on the Cdense_thresh criterion, but the experimental results do not distinguish between them. It would be beneficial to present experimental results for both D-NGD-DDS and D-NGD-PCCS, categorized by Cdense_thresh, to justify this distinction.

---

> ### Author Response · Authors · 2025-08-12
>
> We thank the reviewer for their thorough and constructive feedback. We have incorporated clarifications and new experiments into the revised manuscript to address each of the points raised.
>
> ---
>
> ### 1. D-NGD vs. Primal Gauss–Newton
>
> The update steps are **identical** in the primal and dual formulations. As established in **Theorem 3.1**, D-NGD is the dual (KKT) form of Gauss–Newton/LM. Each iteration solves the **same strictly convex quadratic** (for $\lambda > 0$); with the same damping , the step is **unique**, and a dense **Cholesky** solve returns the same $\Delta\theta$ up to round-off.  Any small practical differences reflect finite-precision effects on ill-conditioned systems which are largely mitigated with **LM-style damping**.
>
> ---
>
> ### 2. Memory Complexity Advantage
>
> The memory advantage is a key aspect of our dual formulation. The complexities for the different methods can be broken down as follows:
>
> - **Primal GN**: $\mathcal{O}(n^2)$
> - **Dense Dual Solver (DDS)**: $\mathcal{O}(m^2)$
> - **Iterative PCG Solver**: $\mathcal{O}(m \cdot l)$ (where $l$ is the Nyström rank)
>
> When the number of parameters $n$ is larger than the number of residual points $m$ used in a batch ($n \gg m$), shifting the quadratic memory dependency from $n$ to $m$ is a significant advantage that enables training of much larger models on the same hardware. We have revised the manuscript to address this point in a new subsection discussing time and memory complexity in Section 4.4.
>
> ---
>
> ### 3. Nyström Preconditioner and the Choice of Landmarks ($l$)
> We agree that a discussion on the **$ \ell $** value would be valuable to the manuscript. **$ \ell $** is the Nyström rank of the preconditioner: increasing $ \ell $ captures more of the Gramian’s leading spectrum, improving conditioning and reducing PCG iterations;  In large-$m$ scenarios, the effectiveness of our proposed preconditioner, built via a Nyström approximation, stems from the fact that the PINN Gramian matrix (and the kernel $\mathcal{K}$, since they share the same nonzero eigenvalues) often has a **rapidly decaying eigenspectrum**. This means a low-rank approximation is sufficient to reduce the large eigenvalue spread responsible for ill-conditioning.
>
> **We introduce the following revisions to expand the discussion in the manuscript(highlighted in blue in Appendix A.12):**
> - Plots of the top 5000 eigenvalues of the Gramian at different training stages, showing that the sharp spectral decay becomes less pronounced as training advances.
> - A table reporting the number of PCG iterations to convergence for different landmark sizes ($ \ell $).
>
>
> ---
>
> ### 4. Dense vs. Iterative Solvers and $ \mathrm{Cdense\_thresh} $
>
> The choice between our dense (**D-DNGD**) and iterative (**PCG-DNGD**) solver is a practical one dictated by hardware memory. $\mathrm{Cdense\_thresh}$ represents the value of $m$ above which the $\mathcal{O}(m^2)$ Gramian matrix no longer fits in GPU memory.
>
> - **Dense Dual Solver (D-DNGD)**: Used for $m < \mathrm{Cdense\_thresh}$. It is preferred when feasible because it provides an **exact** solution to the dual linear system in a single step (factorization + solve).
> - **Iterative Solver (PCG-DNGD)**: Used for $m > \mathrm{Cdense\_thresh}$. It is necessary for problems with a large number of residuals, as it avoids the prohibitive memory bottleneck of the dense approach.
>
>
>
>
> We hope to have addressed the reviewer's concerns and remain at their disposal should further clarifications be needed.

---

> > ### Comment · Reviewer_U1wM · 2025-08-18
> >
> > Thank you for your detailed and thoughtful responses. I appreciate the effort you have put into addressing the questions and concerns I raised. I hope that my comments and suggestions have contributed to further strengthening the quality and clarity of your work.

---

### Review · Reviewer_ttkY · 2025-07-31

**Summary Of Contributions:**

This paper proposes a gradient descent method for minimizing physics-informed losses, called the dual natural gradient descent. The method builds on a previously introduced Gauss-Newton Natural Gradient (GNNG), but proposes to solve a dual problem. The dual problem's Gramian has matrix size that scales with the number of collocation points (spatiotemporal collocation points) rather than the number of parameters in the neural network model. Assuming the former is significantly smaller than the latter, the cost of computing the dual problem is more efficient. The new method is named D-NGD and is able to achieve L2 relative error below 1e-4 for benchmark problems.

**Audience:**

Yes

**Broader Impact Concerns:**

None.

**Claims And Evidence:**

Yes

**Requested Changes:**

- A performance comparison with self-similar BFGS and Broyden methods for the set of experiments in the manuscript would be illuminating.

- A discussion regarding $n$ vs. $m$ and when the scaling of this method becomes favorable in terms of the wall-time. E.g. repeat of the experiments with different values of $n$ and $m$ would be helpful.

**Strengths And Weaknesses:**

Strengths

- The manuscript is clearly written, and the proposed advantage of using the dual problem when the batch-output dimension is much smaller than the NN parameter dimension appears sound.
- The benchmark problems are quite comprehensive, highlighting an overall improvement of the method over standard gradient descent methods.

Weaknesses

- A comparison with a recent work claiming similar level of relative accuracy would be illuminating: Elham Kiyani, Khemraj Shukla, Jorge F. Urbán, Jérôme Darbon, George Em Karniadakis, Which Optimizer Works Best for Physics-Informed Neural Networks and Kolmogorov-Arnold Networks? (https://arxiv.org/abs/2501.16371). They propose a method called self-similar BFGS and Broyden methods. Given the good performance of the methods in the manuscript, comparison showing similar performance would bolster the results in the paper.

- The authors should discuss why they assume the number of collocation points (or the output dimension $m$) should be much less than the number of parameters. Although this is a familiar setting in generic deep learning settings, in the case of PINNs, I am uncertain if the number of parameters should be much more than the number of collocation points.

---

> ### Author Response · Authors · 2025-08-12
>
> **(1) Self-scaled BFGS/Broyden**
> Thank you for the suggestion. The recent work by Kiyani et al. [1] propose self-scaled BFGS/Broyden updates that greatly improve the convergence of Quasi-Newton methods; to the best of our knowledge, these are parameter-space, full-memory methods whose per-step compute and storage grow at least quadratically with the number of parameters $n$. We are not aware of a limited-memory self-scaled variant. Such methods are therefore most suitable for **small networks** and would struggle in most of the experiments we have proposed; in that regime, Gauss–Newton/NGD baselines that already report strong results (e.g., **Jnini et al., 2023** [2]; **Müller and Zeinhofer, 2024** [3]) are the natural comparators. Our main contribution is to scale the GNNG step with linear cost in $n$. Furthermore, we are not aware of a public implementation of the self-scaled variants; combined with their full-memory form, a fair and faithful benchmark at our scales is not feasible within the discussion window.
>
> ---
>
> **(2) $m$ vs. $n$: On the size of the networks and the number of collocation points**
> While both settings have been explored for having large $m$ and/or large $n$, we target regimes where the network size is much larger than batch size, which have been the bottleneck in scaling Natural Gradient Methods to complex and high-dimensional problems.
>
> - **Error-bound view.** Theory for PINNs splits total error into (i) an approximation/optimization part that depends on model capacity (width/depth, hence $n$) and (ii) a sampling part that typically shrinks sublinearly with the number of collocation points $m$ (often about $m^{-1/2}$ for Monte Carlo sampling) [5].
>
> - **Training guidance for PINNs** (Wang et al. [4]) reports that moving to widths around **128–256+** reliably helps on challenging problems and states that networks that are too narrow and shallow lack the expressive capacity to capture complex nonlinear functions. Furthermore, in high input dimension $d$, the needed width tends to grow, and the parameter count scales with both $d$ and depth—so high-dimensional problems naturally imply large $n$. Our study focuses on the large-$n$, small to moderate-$m$ regime.
>
> ---
>
> **Manuscript modifications**
> We highlight the modifications in blue in the manuscript:
>
> - Time & wall-time ($m$ residuals, $n$ parameters): added a discussion in Section 4.4 on time and space complexity.
> - Added a Navier–Stokes $m \times n$ sweep reporting **average per-step cost over 500 iterations** for the dense primal (parameter-space) GNNG vs. dual D-NGD, with a regime map in Appendix A.11.
>
> ---
>
>
> We hope to have addressed the reviewer's concerns and remain at their disposal should further clarifications be needed.
>
>
> **References**
> [1] E. Kiyani, K. Shukla, J. F. Urbán, J. Darbon, G. E. Karniadakis, “Which Optimizer Works Best for Physics-Informed Neural Networks and Kolmogorov–Arnold Networks?”, **2025**, arXiv:2501.16371.
> [2] A. Jnini, F. Vella, M. Zeinhofer, “Gauss-Newton Natural Gradient Descent for Physics-Informed CFD”, **2024**, arXiv:2402.10680.
> [3] J. Müller, M. Zeinhofer, “Achieving High Accuracy with PINNs via Energy Natural Gradients”, **ICML 2023**, arXiv:2302.13163.
> [4] S. Wang, S. Sankaran, H. Wang, P. Perdikaris, “An Expert’s Guide to Training Physics-Informed Neural Networks”, **2023**, arXiv:2308.08468.
> [5] S. Mishra, R. Molinaro, “Estimates on the Generalization Error of Physics-Informed Neural Networks (PINNs) for Approximating PDEs”, **IMA J. Numer. Anal., 2023**, arXiv:2006.16144.

---

> > ### Comment · Reviewer_ttkY · 2025-08-26
> >
> > Thanks to the authors. I would have liked to have seen some comparisons with SS-Broyden regardless, however I understand that can be difficult without publicly available code. I consider my concerns addressed.

---

### Decision · Action_Editor_U6pB · 2025-09-16

**Recommendation:** Accept with minor revision

**Additional Comments:**

A reviewer requested a comparison with SS-Broyden. I agree that some more comparisons would be useful to further validate the method and, while not indispensable, if the authors were to add further comparisons it would strengthen the paper.

Some more discussion to place the approach within the wider ecosystem would be useful: the authors cite Neural Operators in the beginning of the introduction but do not actually mention them anywhere in the text. I would be useful to relate PINNs, and more specifically the proposed approach to other methods, at least to give context to the reader for instance on the difference between learning a solution to an instance vs. operator learning and how the proposed approach would be applied to both, especially in the context of physics informed neural operators which also incorporate a PDE loss.

I was also surprised to not find more details on the choice of architecture and the interplay between the learning algorithm used and the architectural choices. At least some justifications/references to justify the choices made would be useful.

**Audience:**

Yes

**Audience Explanation:**

PINNs are an active area of research and as such this new optimization technique will be of interest to researchers both in that field and interested in training surrogate models for specific PDE problem instances.

**Claims And Evidence:**

Yes

**Claims Explanation:**

All reviewers found the method sound and the claims well supported by theory and experiments, while the paper was found to be clear and easy to follow.